# Inferring differential subcellular localisation in comparative spatial proteomics using BANDLE

Oliver M. Crook [1,2,3] ✉, Colin T. R. Davies [1,3,4], Lisa M. Breckels [1,3], Josie A. Christopher [1,3], Laurent Gatto [5], Paul D. W. Kirk [2,6] & Kathryn S. Lilley [1,3] ✉

The steady-state localisation of proteins provides vital insight into their function. These localisations are context specific with proteins translocating between different subcellular niches upon perturbation of the subcellular environment. Differential localisation, that is a change in the steady-state subcellular location of a protein, provides a step towards mechanistic insight of subcellular protein dynamics. High-accuracy high-throughput mass spectrometry-based methods now exist to map the steady-state localisation and re-localisation of proteins. Here, we describe a principled Bayesian approach, BANDLE, that uses these data to compute the probability that a protein differentially localises upon cellular perturbation. Extensive simulation studies demonstrate that BANDLE reduces the number of both type I and type II errors compared to existing approaches. Application of BANDLE to several datasets recovers well-studied translocations. In an application to cytomegalovirus infection, we obtain insights into the rewiring of the host proteome. Integration of other high-throughput datasets allows us to provide the functional context of these data.

The cell is compartmentalised into organelles and subcellular niches, allowing many biological processes to occur in synchrony[1]. Proteins are localised to these niches in accordance to their function and thus to shed light on the function of a protein, it is necessary to determine its subcellular location. A number of pathologies have implicated incorrect localisation as a contributing factor, including obesity[2], cancers[3], neurological disorders[4], as well as multiple others[5]. It is estimated that up to 50% of proteins reside in multiple locations[6,7], which complicates the study of their localisations. Community approaches have led to substantial improvements in our understanding of subcellular localisation[7,8]. However, image-based approaches are often low in

throughput and high-throughput alternatives are desirable. Furthermore, many biological processes are regulated by re-localisation of proteins, such as transcription factors shuttling from the cytoplasm to the nucleus, which are difficult to map using imaging methods at scale[9].

To simultaneously study the steady-state localisation and re-localisation of proteins, one approach is to couple gentle cell lysis and whole-cell fractionation with high-accuracy mass spectrometry (MS)[6,10–12]. These approaches have already led to high-resolution sub-cellular maps of mouse embryonic stem cell (mESC)[6], human cell lines[11,12], *S. cerevisiae* (bakers' yeast)[13], cyanobacterium[14] and the

[1]Cambridge Centre for Proteomics, Department of Biochemistry, University of Cambridge, CB2 1GA Cambridge, UK. [2]MRC Biostatistics Unit, School of Clinical Medicine, University of Cambridge, Cambridge, UK. [3]Milner Therapeutics Institute, Jeffrey Cheah Biomedical Centre, University of Cambridge, Cambridge CB2 0AW, UK. [4]Mechanistic Biology and Profiling, Discovery Sciences, R&D, AstraZeneca, Cambridge, UK. [5]de Duve Institute, Université catholique de Louvain, Avenue Hippocrate 75, 1200 Brussels, Belgium. [6]Cambridge Institute of Therapeutic Immunology & Infectious Disease (CITIID), Jeffrey Cheah Biomedical Centre, Cambridge Biomedical Campus, University of Cambridge, Cambridge, UK. ✉e-mail: oliver.crook@stats.ox.ac.uk; k.s.lilley@bioc.cam.ac.uk

apicomplexan *Toxoplasma Gondii*[15]. Dynamic experiments have given us unprecedented insight into HCMV infection[16], EGF stimulation[17], and EGFR inhibition[12]. In addition, CRISPR-Cas9 knockouts coupled with spatial proteomics has given insights into AP-4 vesicles[4], as well as AP-5 cargo[18]. These adaptor protein complexes are involved in facilitating the transport of cargo proteins between membranous organelles with both AP-4 and AP-5 associated mutations implicated in severe neurological disease. In a study by Shin et al.[19], the golgin long coiled-coil proteins that selectively capture vesicles destined for the Golgi were re-located to the mitochondria by replacing their Golgi targeting domains with a mitochondrial transmembrane domain[19]. This allowed the authors to readily observe the vesicle cargo and regulatory proteins that are redirected to the mitochondria, while avoiding technical issues that arise because of the redundancy of the golgins and their transient interaction with vesicles. Together, these collections of experiments suggest spatial proteomics can provide unprecedented insight into biological function.

Mass spectrometry-based spatial proteomics currently relies on supervised machine learning methods, such as support vector machines, to assign proteins to a subcellular niche using marker proteins with known localisations[20,21]. Advanced computational approaches have also been developed, including novelty detection algorithms[22,23] and transfer learning approaches[24]. These approaches are implemented in the pRoloc software suite[25,26], which builds on the MSnbase software[27] as part of the Bioconductor project[28,29]. However, most machine learning methods fail to quantify uncertainty (estimate reliability) in the assignment of a protein to an organelle, which is paramount to obtaining a rich interrogation of the data.[30] developed a Bayesian model to analyse spatial proteomics data and highlighted that uncertainty quantification can give insights into patterns of multi-localisation. The method is implemented as a tool as part of the Bioconductor project[31].

In dynamic and comparative experiments; that is, those where we expect re-localisation upon some stimulus to subcellular environment, the data analysis is more challenging. The task can no longer be phrased as a supervised learning problem, but the question under consideration is clear: which proteins have different subcellular niches after cellular perturbation? Procedures to answer this question have been presented by authors[16,17,32,33] and reviewed in ref. 34. The approach of refs. 17, 32 relies on coupling a multivariate outlier test and a reproducibility score—termed the movement-reproducibility (MR) method. A threshold is then applied to these scores to obtain a list of proteins that re-locate; "moving" proteins. However, these scores can be challenging to interpret, since their ranges differ from one experiment to another and require additional replicates to calibrate the scores. Furthermore, the test ignores the spatial context of each protein, rendering the approach inefficient with some applications allowing false discovery rates of up to 23%[18]. Finally, the approach does not quantify uncertainty which is of clear importance when absolute purification of subcellular niches is impossible and multi-localising proteins are present. Recently, Kennedy et al.[33] introduced a computational pipeline for analysing dynamic spatial proteomics experiments by reframing it as a classification task. However, this formulation ignores that some changes in localisation might be shifts in multi-localisation patterns or only partial changes, and it relies on the success of the classification. Furthermore, their approach cannot be applied to replicated experiments and so its applicability is limited. In addition, the authors found that they needed to combine several of the organelle classes together to obtain good results. Finally, the framing of the problem as a classification task only allows a descriptive analysis of the data. These considerations motivate the development of a more sophisticated and reasoned methodology.

We present Bayesian ANalysis of Differential Localisation Experiments (BANDLE)—an integrative semi-supervised functional mixture model, to obtain the probability of a protein being differentially localised between two conditions. Posterior Bayesian computations are performed using Markov-chain Monte-Carlo and so uncertainty estimates are also available[35]. We associate the term *differentially localised* to those proteins which are assigned different subcellular localisations between two conditions. Then, we refer precisely to this phenomenon as differential localisation, throughout the text. Hence, our main quantity of interest is the probability that a protein is differentially localised between two conditions.

BANDLE models the quantitative protein profiles of each subcellular niche in each replicate of each experiment[36]. A first layer of integration combines replicate information in each experiment to obtain the localisation of proteins within a single experimental condition. To probabilistically integrate the two conditions, we use a probability distribution that combines localisation information in each condition so that the datasets are modelled together. By examining the differences in localisations, we can obtain a differential localisation probability. Two prior distributions are proposed: one using a matrix extension of the Dirichlet distribution and another, more flexible prior, based on Pólya-Gamma augmentation[37–39].

In this work, we demonstrate the utility of BANDLE, by first performing extensive simulations and compare to the MR approach. Our simulation study shows that our approach reduces the number of Type I and Type II errors, and, as a result, can report an increased number of differentially localised proteins. These simulations also highlight the robustness of our approach to a number of experimental scenarios including batch effects. Our simulation studies also highlight that BANDLE provides interpretative improvements and clearer visualisations, and makes less restrictive statistical assumptions. We then apply our method to a number of datasets with well-studied examples of differential localisation, including EGF stimulation and AP-4-dependent localisation. We recover known biology and provided additional cases of differential localisation, and suggest that TMEM199, a transmembrane protein involved in Golgi homoeostasis, localisation is potentially AP-4 dependent. Finally, we apply BANDLE to a human cytomegalovirus (HCMV) dataset—a case where MR approach is not applicable because the MR approach requires multiple replicates. Integration of high-throughput transcriptomic and proteomic data, along with degradation assays, acetylation experiments and a cytomegalovirus interactome allows us to provide the functional context of these data. In particular, we provide the spatial context of the HCMV interactome data.

## Results

### The BANDLE workflow

To perform statistical analysis of differential subcellular localisation experiments we developed BANDLE. BANDLE is a semi-supervised integrative functional mixture model that allows the computation of a differential localisation probability. The BANDLE workflow, visualised in Fig. 1, begins with a mass-spectrometry-based spatial proteomics experiment. A cellular perturbation of interest is performed alongside control experiments in wild-type cells or another suitable control, depending on the application. To avoid confounding factors, control and treatment experiments should be performed in parallel with identical mass-spectrometry settings[20,21]. We further recommend checking that the experiment is successful by performing western blots on organelle markers prior to mass-spectrometry analysis, as well as examining the quality of clustering of marker proteins computationally[20,21,40]. Our Bayesian approach, BANDLE, is applicable to experiments with any number of replicates, as well as several subcellular fractionation approaches or mass-spectrometry methods. BANDLE supports multiple mass-spectrometry-based methods, including label-free and labelled quantitation (e.g. SILAC and isobaric tags), and data-dependent and data-independent acquisition. BANDLE models the mass-spectrometry profiles of the subcellular niches using a model that learns spatial correlations known as a Gaussian random

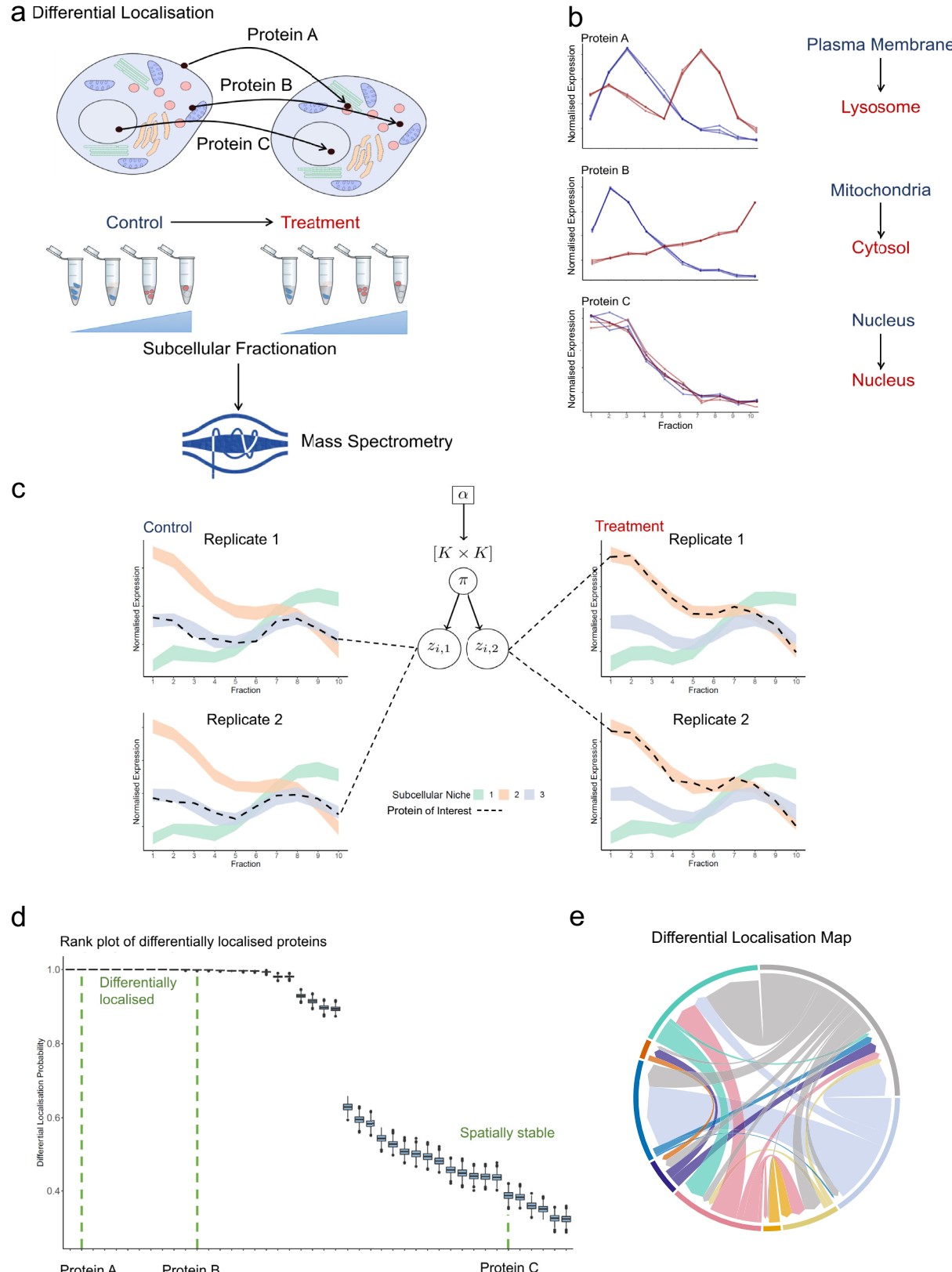

field, which is also frequently referred to as a Gaussian Process (see 'Methods'). Niches are modelled separately across replicates and conditions to allow for variations. Each dataset is hence modelled as a mixture of the different subcellular niches. A distribution of localisations is determined for each protein in each condition. Subsequently, this information is shared across the two conditions by using a

probability distribution that integrates protein localisation information between the datasets. A visualisation in the case of two replicates and two conditions is given in Fig. 1c. By examining the inferred subcellular localisation across the two conditions and using Bayesian inference, we can compute a differential localisation probability. To apply the Bayesian model, we first calibrate the prior based on prior

**Fig. 1 | An overview of the BANDLE workflow. a** A differential localisation experiment is set-up with a perturbation/treatment of interest. Orbitrap Image was generated by Fredrik Edfors at the Noun Project. **b** Mass-spectrometry-based spatial proteomics methods are applied to generate abundance profiles across the subcellular fraction. **c** BANDLE is applied by first calibrating the prior. The prior parameter is denoted by $\alpha$. The model is visualised as follows: each dataset is described as a mixture of subcellular niches, modelled as Gaussian random fields. Allocations are obtained for each condition and then integrated between the two conditions, using a joint prior. $\pi$ is a $K \times K$ matrix, where $K$ is the number of subcellular niches analysed, where entry $(j, h)$ is the probability that a protein is localised to niche $j$ in the control and niche $h$ in the treatment. $z_{i,d}$ is a categorical variable denoting the localisation of protein $i$ in condition $d$. **d** The major results of BANDLE are represented in a rank plot. Sample size = 1000. Data are presented as medians with notches representing the interquartile range (IQR), and the whiskers extending to 1.5*IQR. **e** Example circos plot generated from the results of BANDLE. Source data are provided as a Source data file in the Supplementary material.

predictive checks (simulation of data from the prior probability distribution)[41]. In all scenarios, we check the prior expectation of the number of differentially localised proteins and the probability that more than $l$ proteins are differentially localised. These are reported in the supplement. We then proceed with Bayesian parameter inference using Markov-Chain Monte-Carlo (MCMC), which samples from posterior distribution of the model parameters[35], and the checking of convergence. We visualise our results principally using rank plots, where proteins are ranked from those most likely to be differentially localised or not (Fig. 1d).

## Simulations demonstrate superior performance of BANDLE

To assess the performance of BANDLE and the MR approach, we run a number of simulations allowing us to ascertain the difference between each method in scenarios where we know the ground truth. Two variations of the MR approach were proposed and we compare to both (see 'Methods' section 'The movement-reproducibility method'). We first start with a real dataset from Drosophila embryos and simulate replicates, as well as 20 protein re-localisations[42]. To simulate these datasets a bootstrapping approach is used, coupled with additional noise effects. The first simulation uses a simple bootstrapping approach, where a niche-specific noise component is included (see Supplementary Methods). The subsequent simulations start with the basic bootstrapping approach and add additional effects. The second and third simulations add batch effects: random and systematic respectively (see Supplementary Methods). While the fourth simulation generates misaligned features, i.e. subcellular fraction, across datasets by permuting them (fraction swapping)—this models misaligned fractions between replicates (see Supplementary Methods). The final simulation includes both batch effects and feature permutations. The simulations are repeated 10 times, where each time we simulate entirely new datasets and re-localisations—this is repeated for each simulation task. We assess the methods on two metrics—the area under the curve (AUC) of the true positive rate and false positive rate for the detection of differential localised proteins. Furthermore, we determine the number of correctly differentially localised proteins at fixed thresholds (see Supplementary Methods).

Our proposed method, BANDLE, significantly outperforms the MR methods with respect to AUC in all scenarios ($t$-test $p < 0.01$, Fig. 2). Furthermore, it demonstrates that BANDLE is robust to a variety of situations, including batch effects. We used two separate approach to incorporate prior information into BANDLE. The performance of BANDLE based on the Dirichlet prior is already very good and thus it is unsurprising that we do not observe any significant improvements in AUC by including prior information on correlations captured by the Pólya-Gamma prior. Additional comparisons are made in the supplement where we make similar observations (see Supplementary Note 2).

The improved AUC, which demonstrates improved control of false positives and increased power, translates into increased discovery of differentially localised proteins. Indeed, BANDLE with the Dirichlet prior discovers around twice as many such re-localising proteins than the MR approach (see Supplementary Figs. 2 and 3). Allowing prior correlations through the Pólya-Gamma prior demonstrates that additional differentially localised proteins are discovered. This is an important reality of those performing comparative and

dynamic spatial proteomics experiments, since the experiments become more worthwhile with additional biological discoveries. In practice, the authors of the MR approach advocate additional replicates to calibrate which thresholds are used to declare a protein differentially localised. This assumes that the perturbation of interest does not have a strong effect on the properties of the subcellular niches, which restricts applicability. In contrast, BANDLE does not need additional mass-spectrometry experiments to calibrate its probabilistic ranking meaning more discoveries are made at lower cost. In the Supplementary Methods 21.19), we demonstrate that the differential localisation probabilities provide estimates of the FDR at a 1% level and at all levels once outliers are filtered.

In the following section, we examine the differences between the approaches in a simulated example. There we focus on the output, interpretation and statistical qualities of each approach, rather than the predictive performance of the methods.

## BANDLE quantifies uncertainty and is straightforward to interpret

In this section, we further explore the application of BANDLE with a Dirichlet prior and the MR approach, focusing on the interpretation and statistical properties of the two methods. Again, we simulate dynamic spatial proteomics data, starting from the Drosophila experiment in the scenario in which the MR method performed best. This is where there are cluster-specific noise distributions but no other effects, such as batch effects, were included (see Fig. 2). Sample PCA plots of the data are presented in Fig. 3a. There is a clear pattern of localisations across the data where proteins with known subcellular localisations are closer to each other. However, the organelle distributions clearly overlap and in some cases are highly dispersed—a representation of the challenges faced in real data. These data are annotated with 11 subcellular niches and 888 proteins are measured across 3 replicates of control and 3 of treatment (totalling 6 experiments). Re-localisations are simulated for 20 proteins.

We first apply the MR method according to the methods in refs. 17, 32. We provide a brief description of the approach with full details in the 'Methods'. To begin, the difference profiles are computed by subtracting the quantitative values for each treatment from each control. Then the squared Mahalanobis distance is computed to the centre of the data and under a Gaussian assumption the null hypothesis is that these distances follow a Chi-squared distribution, ergo a $p$-value is obtained. This process is repeated across the 3 replicates and the largest $p$-value was then cubed and then corrected from multiple hypothesis testing using the Benjamini–Hochberg procedure[43]. A negative $\log_{10}$ transform is then performed to obtain the M-score. To produce the R-score, Pearson correlations are computed between each difference profile for all pairwise combination of difference profiles. The lowest of the three R-scores is reported. The M-score and R-score are plotted against each other (see Fig. 3b) and the proteins with high M-score and high R-score are considered differentially localised.

There are a number of assumptions underlying the MR methodology. Firstly, comparing difference profiles pairwise assumes that the features in both datasets exactly correspond. However, this precludes any stimuli that changes the biochemical properties of the

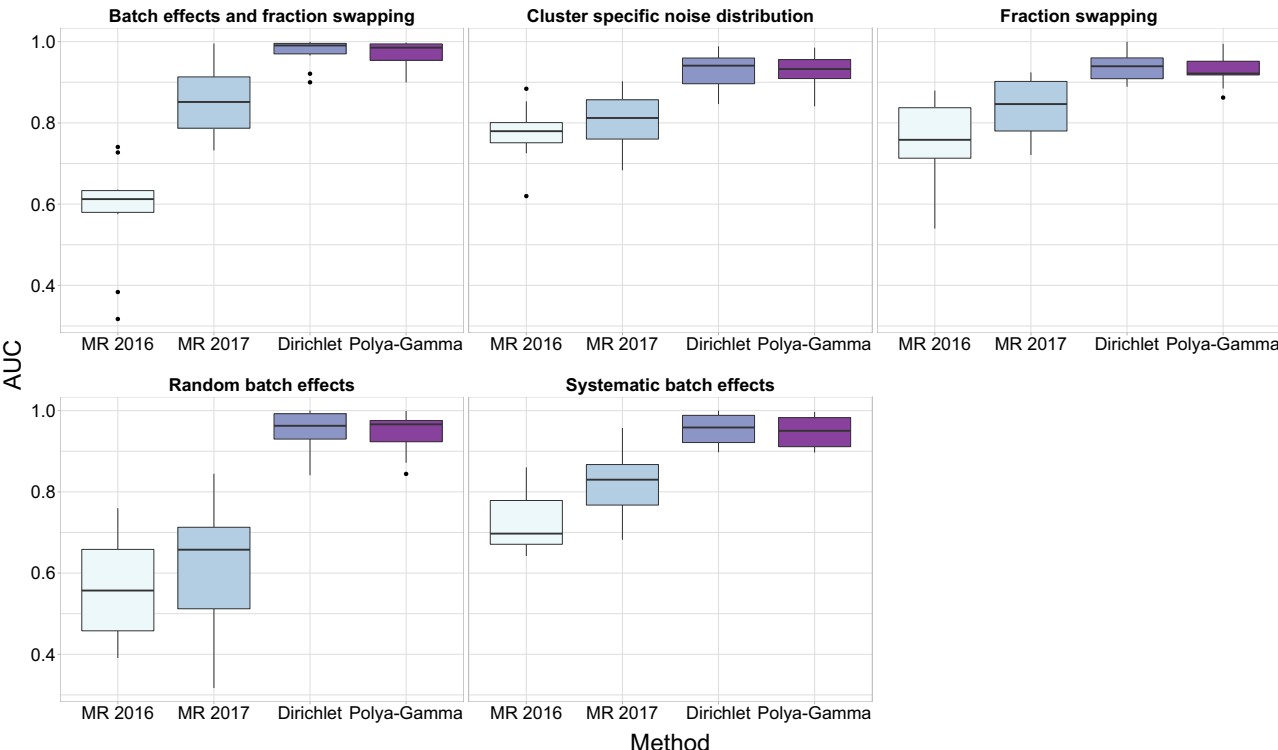

**Fig. 2 | Comparison of BANDLE with other approaches.** Boxplots comparing the performance of the MR approach (2016 and 2017) and our proposed method BANDLE. BANDLE is separated into whether a Dirichlet-based prior was used or if the Polya-Gamma augmentation was applied. Each boxplot corresponds to a different simulation scenario. The boxplots show BANDLE has significantly improved AUC in all scenarios. Simulation are over 10 datasets. Data are presented as medians with notches representing the interquartile range (IQR), and the whiskers extending to 1.5*IQR. Source data are provided as a Source data file in the Supplementary Material.

organelles, since changing these properties may result in differing buoyant densities, pelleting of niches at different centrifugation speeds or differential detergent solubility. Thus, whether density-gradient, differential centrifugation or differential solubilisation is used for organelle separation this assumption must be carefully assessed. Secondly, the Gaussian assumption ignores the natural clustering structure of the data because of the different organelle properties. Indeed, examination of the $p$-value distributions in a histogram (Supplementary Fig. 5) shows that it clearly deviates from the mixture of distributions expected ($p$-values are uniformly distributed under the null). The peaking of $p$-value towards 1 suggests poor distributional assumptions[44]. Thus perhaps the Chi-squared distribution is a poor fit for the statistic of interest. Exploring this further, we fit a Chi-squared and Gamma distribution empirically to the statistics using maximum likelihood estimation (MLE) of the parameters. Figure 3c shows that the Gamma distribution is a better distributional fit—successfully capturing the tail behaviour of the statistic (log Likelihood ratio: 1644 on 1 degree of freedom). The Chi-squared family is nested in the Gamma family of distributions, so if the theoretical Chi-squared distribution was a good fit the distributions would overlap. For a quantitative assessment of model fits we compute the negative log-likelihood of the data given the optimal distributions—the Gamma distribution has a markedly lower negative log-likelihood (log Likelihood Ratio 1644). A $p$-value histogram is provided in Supplementary Note 4. This provides strong evidence that the underlying Gaussian assumptions are likely violated. Thirdly, it is inappropriate to cube $p$-values: to combine $p$-value across experiments one could use Fisher's method[45–47] or the Harmonic mean $p$-value (HMP)[48,49] depending on the context. Indeed, the cube of the $p$-value is no longer a $p$-value. To elaborate, if $\mathcal{P}$ are a set of $p$-values, then under the assumption of the null hypothesis $\mathcal{P}$ is uniformly distributed; however, the cube is clearly not uniformly distributed. Since we no longer work with $p$-values, Benjamini–Hochberg correction becomes meaningless in this context. Transforming these values to a "Movement score", conflates significance with effect size which confounds data interpretation. Finally, summarising to a single pair of scores ignores their variability across experimental replicates.

BANDLE first models each subcellular niche non-parametrically (since the underlying functional forms are unknown[36]). Visualisation of the posterior predictive distributions from these fits for selected subcellular niches is given in Supplementary Fig. 4 (Supplementary Note 3)—we observe a good correspondence between the model and the data. We can see that the different subcellular niches have contrasting correlation structures and thus niche-specific distributions are required. These distributions are specific for each replicate of the experiment and also the two experimental conditions. The information from the replicates, and the control and treatment are combined using an integrative mixture model. Briefly, mixing proportions are defined across datasets allowing information to be shared between the control and treatment (see 'Methods' for more details). This formulation allows us to compute the probability that a protein is assigned to a different subcellular niche between the two experiments—the differential localisation probability. The proteins can then be ranked from most probably differentially localised to least (Fig. 3d). The figure is simple to interpret: the proteins with highest rank are the most likely to have differentially localised between the experiment, having been confidently assigned to different subcellular niches in the control versus treatment. The proteins with lowest rank are highly unlikely to have moved during the experiment—the localisations are stable. This is important information in itself, especially when combined with other information; such as, changes in abundance or post-translational modification. Figure 3d (right) shows the 30 proteins with highest rank

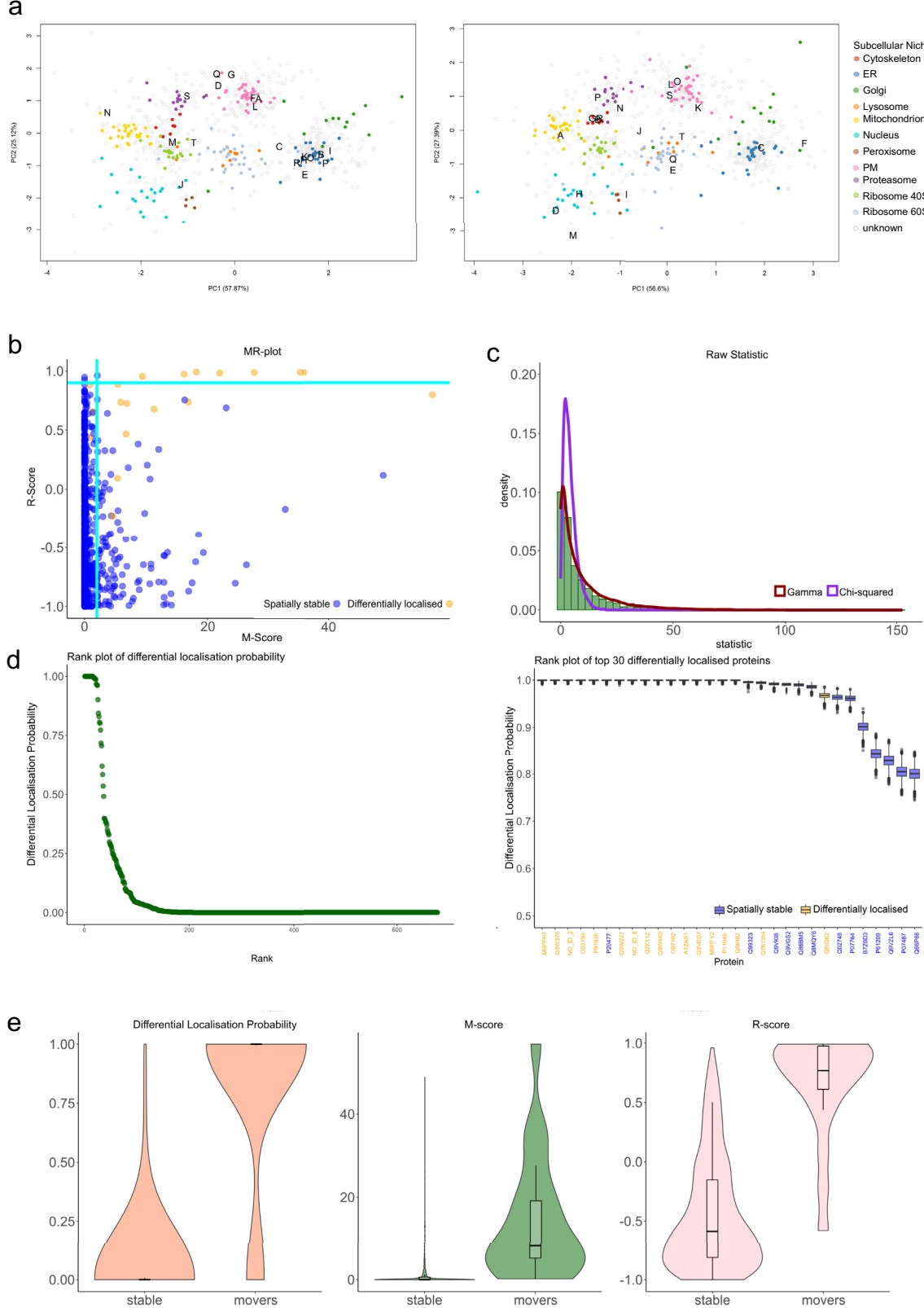

visualising the uncertainty in the differential localisation probability (see 'Methods'). This ranking allows us to prioritise which proteins to follow up in validation experiments. Example changes in localisation are provided in Supplementary Note 5. The ranking can also be mapped onto other experimental data, such as expression or protein–protein interaction data. The probabilistic ranking produced by BANDLE is more closely aligned with the phenomenon of interest. Indeed, if we divide the data into the proteins that were differentially localised and those that were not. Then from plotting the distribution of the statistics from the respective methods, it is clear that output from BANDLE is most closely associated with re-localisation events (Fig. 3e).

**Fig. 3 | A detailed comparison of BANDLE and the movement-reproducibility approach. a** Example PCA plots where pointers correspond to proteins. Marker proteins are coloured according to their subcellular niche, while proteins with unknown localisation are in grey. Simulated translocations are indicated in black letters, where the left corresponds to first control and right to first the perturbed simulated datasets. **b** An MR-plot showing movement score against reproducibility score. Each pointer corresponds to a protein and orange pointers correspond to simulated translocations and blue otherwise. Teal lines are drawn at suggested thresholds with proteins in the top right corner considered hits. **c** A histogram of the raw statistics underlying the MR method. A Chi-square (orange) and Gamma (blue) fit are overlaid (obtained using maximum likelihood estimation). The Gamma distribution clearly captures the tail behaviour. **d** A BANDLE rank plot where proteins are ranked from most to least likely to differentially localised. The differentially localisation probability is recorded on the y-axis. (right) A BANDLE rank plot of the top 30 differentially localised proteins with uncertainty estimates for the differential localisation probability. Proteins marked in orange were simulated translocations. **e** Violin plots for the differential localisation probabilities (BANDLE), the M score (MR method) and R score (MR method). The distributions are split between differential localised (movers) and spatially stable proteins. Clearly, the differential localisation probabilities correlate most closely with the phenomena of interest. Boxplots are presented as medians with notches representing the interquartile range (IQR), and the whiskers extending to 1.5*IQR. Sample size is 888. Source data are provided as a Source data file in the Supplementary Material.

To examine the calibration of the probabilities we computed the expected calibration error—the average difference between the predicted probability and true probability (see Supplementary Methods). We found that the expected calibration error was 0.042, suggesting that these probabilities are well-calibrated, since they are similar to typical errors found on approaches that have undergone post hoc calibration[50]. For further simulations, see the Supplementary Methods. We also performed a prior-sensitivity analysis and found that the influence of the prior is weak (see Supplementary Note 11). We also examine the effect of normalisation procedures on the outcomes of the analysis and found that this has a limited influence on the performance of BANDLE but can influence the results of the MR approach considerably (see Supplementary Methods).

## Characterising differential localisation upon EGF stimulation

Having carefully assessed the statistical properties of our approach, BANDLE, and the MR method, we apply these approaches to a number of datasets. First, we consider the Dynamic Organeller Maps (DOMs) dataset of ref. 17, exploring the effects of EGF stimulation in HeLa cells. In this experiment, SILAC labelled HeLa cells were cultured and recombinant EGF was added to the culture at a concentration of 20 ng ml$^{-1}$ (see ref. 17). A total of 2237 complete protein profiles were measured across 3 replicates of control and 3 replicates of EGF treated HeLa cells. Principal Component Analysis (PCA) projections of the data can be visualised in the supplement (Supplementary Figs. 24 and 25). A quality control assessment was performed using the approach of ref. 40. As a result, nuclear pore complex, peroxisome and Golgi annotations were removed, since the marker proteins of these classes were highly dispersed.

The MR method was applied as described in the 'Methods' and the results can be visualised in Fig. 4a. 7 proteins are predicted to be differentially localised using the MR method with the thresholds suggested by ref. 17. These include 3 core proteins of the EGF signalling pathway SHC1, GRB2 and EGFR[51] and other, potentially related, proteins TMEM214, ACOT2, AHNAK, PKN2. Since the MR approach does not provide information about how the functional residency of the proteins change, it is challenging to interpret these results without further analytical approaches.

To quantify uncertainty and gain deeper insight into the perturbation of HeLa cell after EGF stimulation we applied our BANDLE pipeline, with an informative prior (see 'Methods'). Sensitivity to these prior choices are assessed in the Supplementary Materials (Supplementary Note 11). Firstly, the rank plots display a characteristic shape suggesting that most proteins are unlikely to be differentially localised upon EGF stimulation (Fig. 4b). Furthermore, we provide uncertainty estimates in the probability that a protein is differentially localised for selected top proteins (Fig. 4c). Furthermore, we visualise the change in localisation for the proteins known to re-localise upon EGF stimulation: SHC1, GRB2 and EGFR (Fig. 4e). This is displayed by projecting the posterior localisation probabilities on to the corresponding PCA coordinates. These probabilities are then smoothed using a Nadaraya-Watson kernel estimator[52,53] and visualised as contours. PCA plots of the raw data are found in the Supplementary Materials (Supplementary Note 13).

Given the well-documented interplay between phosphorylation and subcellular localisation[54–57], we hypothesised that proteins with the greatest differential phosphorylation would correlate with proteins that were more likely to be differentially localised. To this end, we integrated our analysis with a time-resolved phosphoproteomic dataset of EGF stimulation using MS-based quantitation[58]. In their study, cells were harvested at eight different time points after EGF stimulation: 0, 2, 4, 8, 16, 32, 64 and 128 min (see supplementary Note 7). Cells were harvested and protein digested to peptides using trypsin. Peptides corresponding to each time point were labelled with a different iTRAQ tag before combining all samples together and quantifying using LC-MS/MS. Immunoprecipitation was used to enrich for phosphorylated tyrosine residues[59] and the enrichment of phosphosites on serine and threonine residues was performed via immobilized metal affinity chromatography (IMAC)[60,61].

For each phosphopeptide corresponding to a unique protein, we computed the largest $log_2$ fold change observed across the time course. Given that the changes in localisation occur within 20 min, we restricted ourselves to the first 6 time points[17]. We then took the top 10 proteins ranked by each of the MR method and BANDLE. These rankings are then correlated with rankings obtained from the changes in phosphorylation. The Spearman rank correlations were recomputed for 5000 bootstrap resamples to obtain bootstrap distributions of correlations (see Fig. 4). We report the mean correlation and the 95% bootstrap confidence intervals. The correlation between the ranks of the MR method and the phosphoproteomic dataset was $\rho_S = 0.40$ (−0.49, 0.85), while the correlation when using the ranking of BANDLE was $\rho_S = 0.68$ (0.02, 0.98). That is, phosphorylated proteins are more likely to be differential localised and this signal is more clear using the ranking obtained from using BANDLE. Alongside the statistical and interpretable benefits of BANDLE, it is clear the approach has the utility to provide insight into localisation dynamics.

## BANDLE obtains deeper insights into AP-4-dependent localisation

The adaptor protein (AP) complexes are a set of heterotetrameric complexes, which transport transmembrane cargo protein vesicles[62]. The AP1-3 complexes are well characterised: AP-1 mediates the transport of lysosomal hydrolases from the trans-Golgi to the endosomes[63,64]; AP-2 has a significant role in the regulation of endocytosis[65]; AP-3 is involved in the sorting of trans-Golgi proteins targeted to the lysosome[66]. The role of the AP-4 complex has become better understood in recent years[4,67–73] and is of noted interest because loss-of-function mutations resulting in early-onset progressive spastic paraplegia[74]. The altered subcellular distribution of ATG9A, as a result of loss-of-functions AP-4 mutation[4,69,70], is believed to be a key contributor to the pathology of AP-4 deficiency syndrome[4,69–73].

AP-4 consists of four subunits ($\beta$4, $\varepsilon$, $\mu$4 and $\sigma$4) forming an obligate complex[66,4] study the functional role of AP-4 using spatial proteomics; in particular, the DOM workflow mentioned previously. As

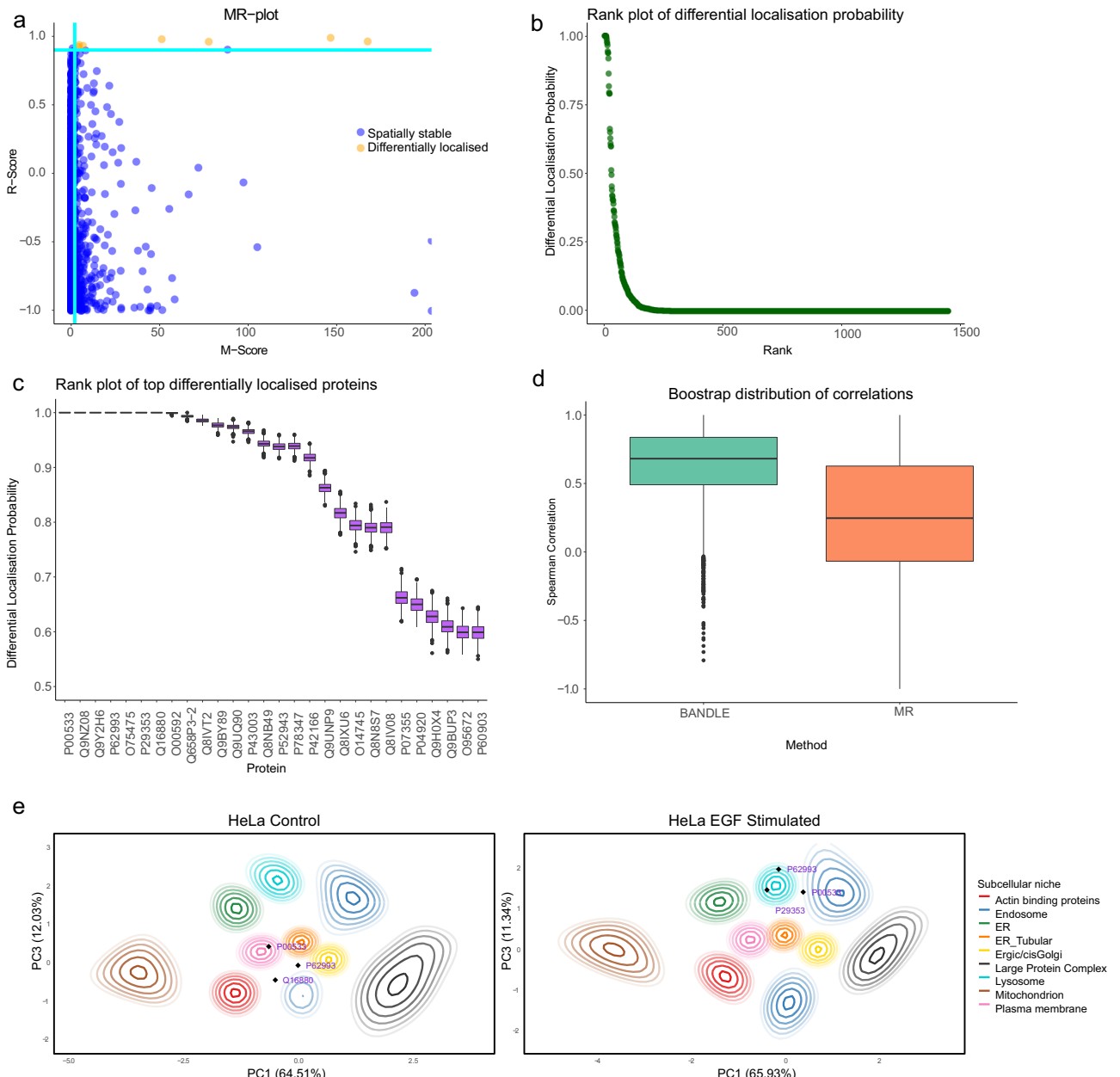

**Fig. 4 | Analysis of an EGF stimulation dataset. a** An MR-plot where dark green lines are drawn at suggested threshold and hits are highlighted in orange. **b** BANDLE rank plot showing the distribution of differentially localised proteins. **c** The top differentially localised proteins from BANDLE plotted with uncertainty estimates. **d** Bootstrap distributions of correlations with a phosphoproteomic time-course experiment. The BANDLE confidence intervals differ significantly from 0, while the MR method do not. 5000 bootstrap samples were used with data presented as medians with notches representing the interquartile range (IQR), and the whiskers extending to 1.5*IQR. **e** PCA plots with (smoothed) localisation probabilities project onto them. Each colour represents an organelle and ellipses represent lines of isoprobability. The inner ellipse corresponds to 0.99 and the proceed line 0.95 with further lines decreasing by 0.05 each time. The annotated proteins demonstrate examples of differential localisations. EGFR (P005330) clearly relocalises from the PM to endosome, while SHC-1 (P29353) and GRB2 (P62993) relocalise from unknown localisation to the lysosome. Source data are provided as a Source Data file in the Supplementary Material.

part of their study, they used AP-4 CRISPR knockout cells to interrogate the effect on the spatial proteome when AP-4 function has been ablated.

Re-analysis of this subcellular proteomics experiment provides full quantitative measurements for 3926 proteins across two replicates of wild-type cells and two replicates where the $\beta4$ subunit has been knocked-out. They also produce two replicates where AP4E1 is knocked-out but this is not considered here for brevity. The data are visualised as PCA plots (see Supplementary Note 14). As in the previous analysis, we run a quality control step removing the actin binding protein and nuclear pore complex annotations[40]. This

dataset is particularly challenging to analyse because there are only two replicates for each condition. The value of Bayesian analysis is the ability to provide prior information to regularise, as well as the quantification of uncertainty, which is more critical in data sparse scenarios.

Previous application of the MR methods led to authors to find that SERINC 1 (Q9NRX5), SERINC 3 (Q13530) were differentially localised, as well as an altered subcellular distribution for ATG9A (Q7Z3C6)[4]. Their results suggest these are cargo proteins of the AP-4 complex that are packaged into vesicles at the trans-Golgi before being transported to the cell periphery. Altogether, their results suggest AP-4 provides

spatial regulation of autophagy and that AP-4 neurological pathology is linked to disturbances in membrane trafficking in neurons[4,69].

We apply our method BANDLE to gain further insights into AP-4-dependent localisation. We compute the differential localisation probability; the associated uncertainty estimates and rank proteins according to this statistic (see Fig. 5a and Supplementary Note 9). Characteristic S-shaped plots are observed with most proteins not differentially localised upon knock-out of AP-4 $\beta$4. The results of both SERINC 1 and 3 are validated, as we compute a differential localisation probability greater than 0.95 for these proteins. Furthermore, 16 of the top 20 proteins are membrane-bound or membrane-associated proteins (FDR < 0.01 hyper-geometric test). To demonstrate the benefit of our probabilistic ranking, we perform two-sided KS rank test against the functional annotations provided in the STRING database (corrected for multiple testing within each functional framework)[75]. We find that processes such as ER to Golgi transport and lipid metabolism are more highly ranked than would be expected at random (FDR < 0.01), as well as endosomes and Golgi localisations (FDR < 0.01). While processes associated with translation, ribosome localisation and function appear significantly lower in the ranking (FDR < 0.01). As expected, this provides a high-level overview and evidence for the functional nature of AP-4 in the secretory pathway.

Taking a more precise view on our results, we examine the top 20 differentially localised proteins in more detail. We compute the Spearman correlation matrix between these proteins and observe strong correlation, suggesting the proteins act in a coordinated way (see Fig. 5b). Visualising the data in a heatmap (Fig. 5b), after mean and variance normalisation, we observe a highly concordant pattern: most proteins are enriched in fractions 4 and 5. These fractions are obtained from the highest centrifugation speeds and so differentially pellet light membrane organelles, such as endosomes and lysosomes[11,17]. Again, further evidence for the role of AP-4-dependent localisation dynamics within the secretary pathway.

In Fig. 5b, we observed a large cluster of 9 proteins, which included SERINC 1 and 3. Amongst these 9 proteins is SLC38A2, a ubiquitously expressed amino-acid transporter that is widely expressed in the central nervous system and is recruited to the plasma membrane from a pool localised in the trans-Golgi[76–79]. Thus, its suggested differential localisation here provides further evidence for the role of AP-4 as a membrane trafficker from the trans-Golgi. Another protein in this cluster is TMEM 199 (Q8N511) a protein of unknown function that is involved in lysosomal degradation[80]. Furthermore, it has been implicated in Golgi homoeostasis but the functional nature of this process is unknown[81]. Probing further, we observe that TMEM199 acts in a coordinated fashion with SERINC 1 and 3. Marked re-localisations are observed on PCA plots toward the endo/lysosomal regions (see Fig. 5c) and we note that the quantitative profiles of SERINC 1, SERINC 3 and TMEM199 act in an analogous way upon AP-4 knockout (see Fig. 5d). Our findings motivate additional studies to elucidate AP-4 dependent localisation and separate these observations from potential clonal artefacts.

## Rewiring the proteome under cytomegalovirus infection

Human cytomegalovirus (HCMV) infection is a ubiquitous herpesvirus that burdens the majority of the population[82]. In healthy immune systems, HCMV establishes latent infection following initial viral communication[83] and reactivation can lead to serious pathology in some imunno-compromised individuals[84]. HCMV has the largest genome of any known human virus, at 236 kbp it encodes for over 170 proteins that modulate almost all aspects of the hosts cellular environment for its benefit[85–87].

Initial viral infection involves endocytosis of the virion into the cell[88], host machinery is then used to transport viral capsids into the nucleus[89]. Within the host nucleus viral transcription and genome replication occurs[90–92]. Meanwhile, other viral proteins are targeted to

the secretory pathway to inhibit the host immune response and regulate the expression of viral genes[93–98], rewire signalling pathways[99] and modulate metabolism[100]. In later phases, the cellular trafficking pathways and the secretory organelles are hijacked for the formation of the viral assembly complex (vAC)[101–105]. Due to the diversity of cellular processes manipulated during HCMV infection, it is often used as a paradigm to analyse virus-host interactions[106].

There has been a recent flurry in applying system-wide proteomic approaches to the HCMV infection model.[106] developed quantitative temporal viromics a multiplexed proteomic approach to understand the temporal response of thousands of cellular host and viral proteins. More recently, to discover proteins involved in the innate immune response, a multiplexed proteasome-lysosome degradation assay found that more than 100 host proteins are degraded shortly after viral-infection[107]. Meanwhile, a comprehensive mass spectrometry interactome analysis has identified thousands of host-virus interactions[108]. Furthermore, high-throughput temporal proteomic analysis has revealed the importance of protein acetylation (post-translational modification of lysine amino acids), as an integral component during HCMV infection[109].

Reference 16 use spatial and temporal proteomics to investigate the response of the human host proteome to HCMV infection. The authors perform subcellular fractionation on uninfected (control) and HCMV-infected (treated) cells at 5 different time points: 24, 48, 72, 96, 120 h post infection (hpi). The authors then used neural networks to classify proteins to subcellular niches at each time point in the control and treated cells, allowing a descriptive initial analysis of the data. Proteins with differential classification at each time point are those that are believed to be differential localised. However, the challenge of this study is that only a single replicate is produced in each condition. This renders the MR method of ref. 17 inapplicable.

Differential classification is a reasonable approach to probe differential localisation though it neglects information shared across both experiments and it is not quantitative (i.e. no $p$-value or posterior probability of change). In the case of single replicates, by sharing information and providing prior information we are able to improve inference and obtain deeper insights. We apply BANDLE to control and HCMV-treated cells at 24 hpi, in the interest of brevity, to explore further the host spatial-temporal proteome (see Supplementary Notes 15 and 20). Our analysis reflects a potential rewiring of the proteome with many possible proteins differentially localised on HCMV infection. We highlight an example of differential localisation with SCARB1 (see Fig. 6a), with a localisation in the secretory pathway shifting toward a PM/cytosolic localisation, similar to what has previously been observed[16].

To obtain global insights into the functional behaviour of the differentially localised proteins, we performed a Gene Ontology (GO) enrichment analysis. An extensive list of terms is enriched and these can be divided broadly into subcategories such as translation and transcription; transport; viral processes; and immune process (see Supplementary Note 16). These results reflect closely the early phase of HCMV infection[87]. Pathway enrichment analysis highlights terms related to a viral infection (Viral mRNA Translation, Influenza Life Cycle, Infectious disease, Innate Immune System, Immune System, MHC class II antigen presentation, Antigen processing-Cross presentation, Host Interactions of HIV factors, HIV Infection) (see Supplementary Note 16). Pathway analysis also reveals known processes that are modulated during HCMV infection, such as membrane trafficking[110–112], Extracellular matrix organization[113] and Rab regulation of trafficking[114].

## Integrating HCMV proteomic datasets to add functional relevance to spatial proteomics data

The spatial information obtained here allows us to perform careful integration with other high-resolution proteomic datasets. The degradation screens by ref. 107 identified proteins that were actively

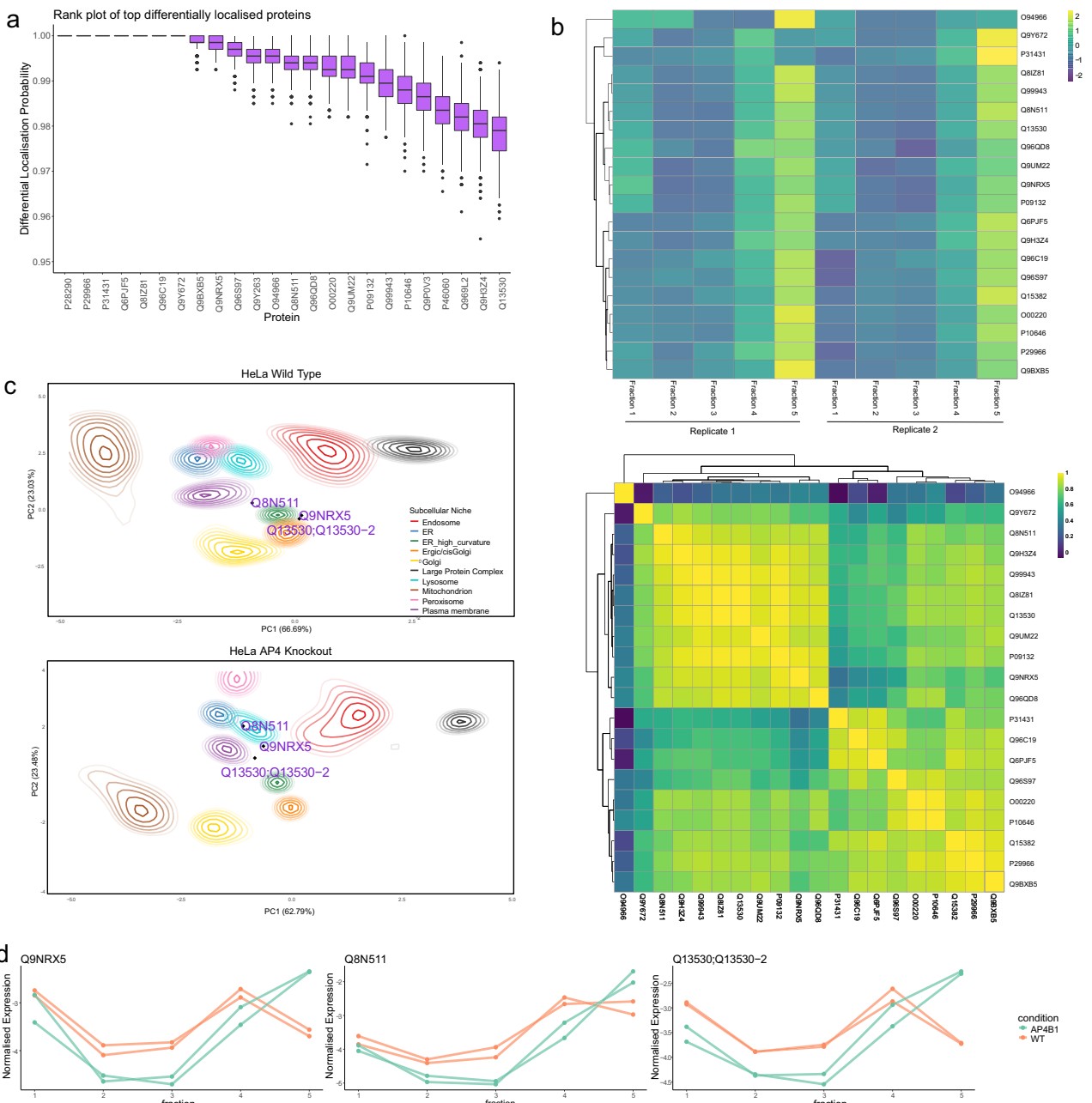

**Fig. 5 | BANDLE applied to the AP-4 dataset. a** The top differentially localised proteins from BANDLE plotted with uncertainty estimates. Data are presented as medians with notches representing the interquartile range (IQR), and the whiskers extending to 1.5*IQR. 667 MCMC samples were used. **b** A Spearman correlation heatmap showing strong correlation behaviours of proteins that have AP-4 dependent localisation (lower). Normalised mass-spectrometry profiles plotted as a heatmap from the AP-4 knockout data. Proteins are shown to have similar behaviour with greater intensity in fraction 5, where light membrane organelles are likely

to pellet (upper). **c** PCA plots with (smoothed) localisation probabilities project onto them. Each colour represents an organelle and ellipses represent lines of isoprobability. The inner ellipse corresponds to 0.99 and the proceed line 0.95 with subsequent lines decreasing by 0.05 each time. The proteins SERINC 1 and 3, as well as TMEM199 are highlighted demonstrating example relocalisations. **d** Normalised abundance profiles showing that SERINC 1, SERINC 3 and TMEM199 show similar behaviour upon knockout of AP-4. Source data are provided as a Source data file in the Supplementary Material.

degraded during HCMV infection but gave no information regarding the spatial location of the targets. To determine the location of host proteins targeted by HCMV for degradation, the BANDLE revised spatial data at 24 hpi was overlapped with proteins that were degraded by the proteasome or lysosome. The subcellular location of the host proteins is displayed for the 24 h timepoint. To determine the spatial granularity of the degradation data we tested whether the proteins assigned to each spatial pattern had a significantly different degradation distribution than the distribution of all proteins in the experiment (*t*-test). We note that proteins that are differentially localised are no

more likely to be targeted for degradation than those that are not (see Supplementary Note 17).

Analysis of changes in protein abundance can be used to generate turnover rates in both HCMV-infected and mock-infected cells. Comparing these turnover rates allows us to calculate the rescue ratio, which identifies proteins that exhibit increased degradation during viral infection compared to baseline. Specifically, the rescue ratio is obtained by comparing abundance during HCMV infection ∓ inhibitor with protein abundance during mock infection ∓ inhibitor. Degradation data from ref. 107 are overlaid as a heatmap, showing a

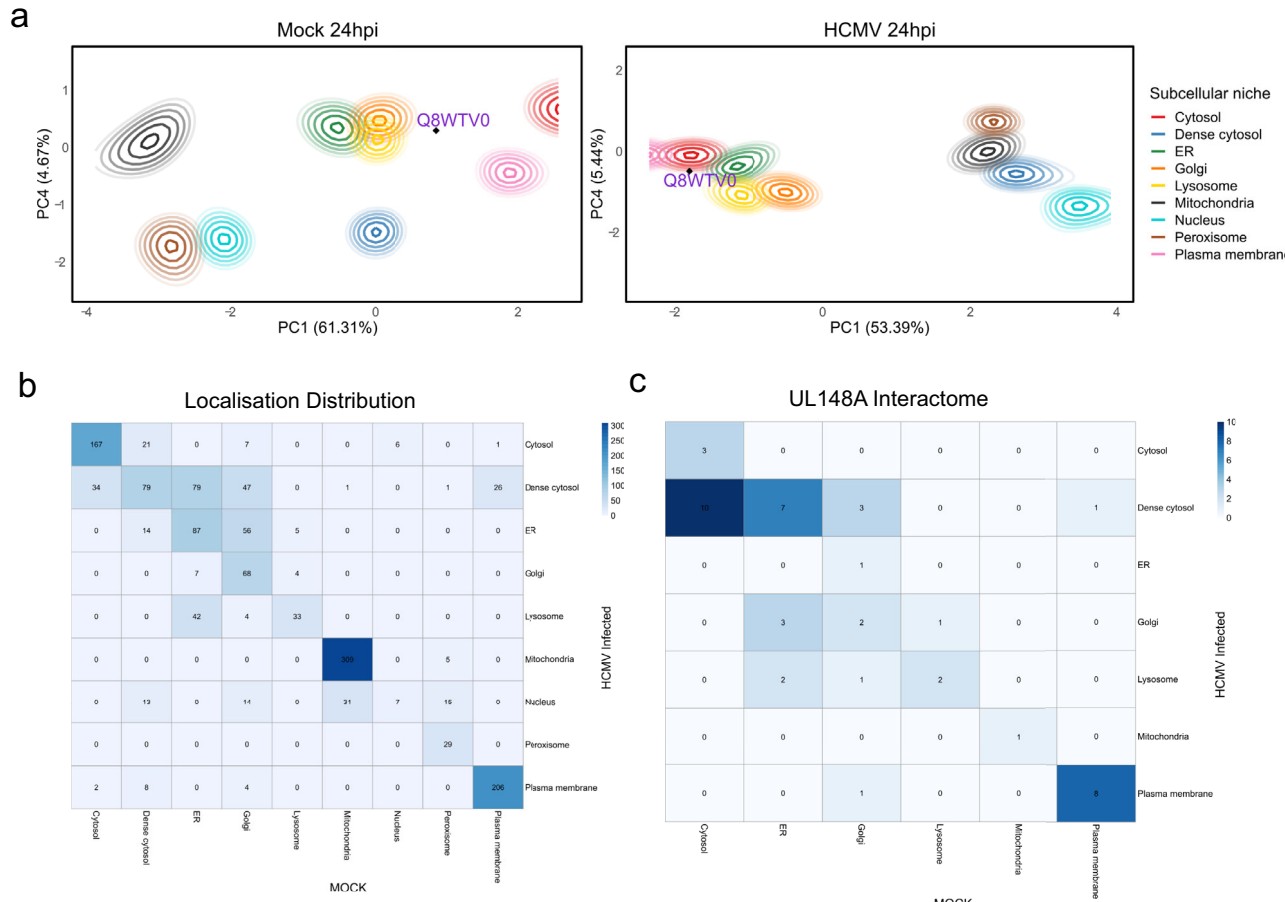

**Fig. 6 | BANDLE applied to the HCMV dataset. a** PCA plots with (smoothed) localisation probabilities projected onto them. Each colour represents a subcellular niche and ellipses represent lines of isoprobability. The inner ellipse corresponds to 0.99 and the proceed line 0.95 with further lines decreasing by 0.05 each time. The relocalisation of SCARB1 is highlighted on the plot. **b** The spatial allocation derived from BANDLE where each entry of the heatmap is the number of proteins. Off-diagonal entries only include confident differential localisations with probability >0.999. **c** UL148A interactome mapped onto the BANDLE determined spatial patterns. Source data are provided as a Source data file in the Supplementary Material.

$-\log_{10}(p$-value) for each inhibitor (see Supplementary Figs. 37 and 39). For proteasomal targeted proteins (MG132 inhibitor), the data highlight a high number of proteins degraded from the mitochondria. The mitochondria act as a signalling platform for apoptosis and innate immunity and it is already well-established that HCMV can subvert these processes to its advantage[115]. Furthermore, there is a high degree of protein degradation as one might expect in proteasome fractions (dense cytosol), with an enrichment of proteins recruited from the ER and cytosol (see Supplementary Fig. 37). For lysosomal targeted proteins (leupeptin) there was a high degree of proteins degraded from the mitochondria, cytosol and plasma membrane. There were also several proteins degraded that moved from the cytosol to the dense cytosol (see Supplementary Note 17).

Many host proteins are up-or-down regulated upon HCMV infection[106]. We examine more recent abundance data from ref. 109 at 24 hpi and first we note that differentially localised proteins are not more abundant than spatially stable proteins (see Supplementary Fig. 43). However, we see a strong spatial pattern when we overlay the abundance pattern on a heatmap. In Supplementary Fig. 36, we report the mean log2 fold change for proteins stratified according to predicted subcellular localisation. It is important to combine spatial and abundance data, since a differentially localised protein may not undergo a true translocation event but rather a new pool of proteins is synthesised. Which of these options is true could be further investigated by coupling spatial proteomics workflows with time-resolved incorporation of stable isotope labelled amino acids or azido-homoalanine to interrogate the location of newly synthesized proteins. The significance of these abundance changes is highlighted in Supplementary Fig. 35. For example, there is a significant decrease in the abundance of the protein recruited to the dense cytosol from the ER (see Supplementary Fig. 44). Some of the larger changes are not significant because there are too few proteins with the same spatial pattern. We note that FAM3C, a protein involved in platelet degranulation, is upregulated at 24 hpi. Furthermore, FAM3C relocalises from the Golgi to the lysosome, its Golgi localisation is in concordance with the Human Protein Atlas (HPA)[7] and its lysosome relocalisation suggests that it is trafficked through the secretory pathway before undergoing degranulation.

Upon integration of the acetylation data of ref. 109, the spatial patterns are much more nuanced (see Supplementary Note 18). Perhaps surprisingly, we do not observe increased acetylation levels amongst differentially localised proteins (see Supplementary Fig. 47). The only significant pattern is for proteins relocalising from the dense cytosol to the cytosol; however, we observe this is driven by a single protein Skp1 (see Supplementary Fig. 48), which shows a 2.5-fold increase in acetylation at 24 hpi for Skp1 and there is an increase in its RNA transcript at 24 hpi[107]. The Skp1 protein is part of an E3 ubiquitin ligase complex that targets proteins for degradation. E3 ligases are often manipulated by viruses in order to control cellular processes to create a cell state that benefits viral replication and survival[116]. It is therefore possible that HCMV is controlling Skp1 activity through

acetylation at its C-terminus, leading to its translocation and likely change in function.

The recent publication of the HCMV interactome has provided a wealth of data that gives insights into the function of the 170 canonical and two non-canonical viral protein-coding genes[108]. However, a common difficulty with analysing large interactome projects is the ability to reduce the number of false-positive interactions, leading to poor agreement between experimental and computational datasets. This can be controlled through replicates, supervised machine learning and increased statistical stringency; however, background contamination can never be eliminated. If a protein is located in a single location, you would expect true positive interactors to be located in the same subcellular compartment. Therefore, to narrow the list of viral-protein interactors, we overlapped spatial information from Beltran et al.[16] with the viral interactors from Nobre et al.[108] (Fig. 6c, Supplementary Figs. 51 and 52). This provides a far more stringent set of high confidence protein–protein interactions. Although, this comes at the cost of removing some interactions between proteins that are located in more than one subcellular location and are therefore absent from the spatial dataset from Beltran et al.[16].

We plot a heatmap to indicate the spatial distribution of the host proteins (Fig. 6c). The overall distribution is plotted in the heatmap of Fig. 6b. Firstly, we are interested in scenarios where the interacting host proteins were more likely to retain their localisation upon HCMV infection (than the computed posterior distribution would have predicted). Thus, for each viral bait, we simulated from a binomial $A$ - $Bin(n, p)$ where $p$ is the posterior probability that a random protein was assigned to the same localisation and $n$ is the number of interactors of that viral bait. We then simulated from this distribution 5000 times to obtain a histogram (see Supplementary Note 19). Viral baits of interest are those where the observed statistic was in the tails of these histograms.

Examples of such cases are shown for viral proteins UL8 and UL70 (Supplementary Figs. 52 and 53). The majority of UL8 interactors were located in the plasma membrane and cytosol. UL8 is a transmembrane protein that is transiently localised at the cell surface, with a small cytoplasmic pool[117], perfectly mimicking the location of the majority of UL8 interactors. Practically all UL70 interactors were located in the cytosol. Viral UL70 is a primase known to locate to both the nucleus and cytoplasmic compartments during HCMV infection[118]. As the nucleus was removed prior to fractionation then one expects only to be able to interrogate cytosolic interactors. An example where the host proteins were spatially diffuse was UL148A an elusive viral protein of unknown function, believed to be involved with modulating the innate immune response[119]. UL148A appears to interact with host proteins distributed throughout the cell suggesting it is highly promiscuous (Fig. 6c). Perhaps UL148A is a protein with multiple possible functions[120] making its function hard to pinpoint and such an observation would not be uncommon for viral proteins because of limited genomic size[121,122]. These results illustrate the strength in overlapping spatial proteomics with interactome studies to decrease the number of false positives and focus research on higher confidence protein–protein interactions. The entire list of spatially resolved viral protein interactions is shown in the Supplementary Material.

## Discussion

We have presented a Bayesian model for comparative and dynamic spatial proteomic experiments. Unlike current approaches, our flexible integrative mixture model allows any number of replicate experiments to be included. Furthermore, subcellular profiles are modelled separately for each condition and each replicate, allowing cases where the correlation profiles differ between experiments. Crucially, our model facilitates the computation of differential localisation probability, which cannot be performed by other methods in the literature. Furthermore, BANDLE probabilistically assigns proteins to organelles and can model outliers meaning that further supervised machine learning after application of BANDLE is not required. The probabilistic ranking obtained from BANDLE can be used for downstream pathway or GO enrichment analysis, likewise it can be mapped onto other orthogonal high-throughput datasets.

We compared BANDLE to the MR approach of refs. 17, 32. The MR method is not as broadly applicable as BANDLE, and BANDLE does not require additional experiments to interpret the thresholds. In our careful simulation study, we demonstrate reduced Type 1 error and increased power when using our approach. In a further simulation, we demonstrated that BANDLE has more desirable statistical properties than the MR approach, the results are easier to interpret and more information is available. Since we are in a Bayesian framework, our approach also quantifies uncertainty allowing us to obtain differential localisation probabilities.

Application of our approach to three dynamic and comparative mass-spectrometry-based spatial proteomic experiments demonstrates the broad applicability of our approach. We validated many previously known findings in the literature, placing confidence in these results. When BANDLE was applied to EGF stimulation dataset, we saw increased correlation between our differential localisation results and a phosphoproteomic timecourse than when compared to the results of the MR approach.

We applied BANDLE to an AP-4 knockout dataset to investigate AP-4 dependent localisation and, as with other studies, we observe SERINC 1 and SERINC 3 are examples AP-4 of Cargo. Furthermore, we implicate TMEM199 as potentially overlooked AP-4 cargo, though it remains to rule it out as a potential clonal artefact. We apply BANDLE to a dataset where the MR approach is not applicable–an HCMV infection spatial proteomic dataset. Pathway and GO enrichment results implicate differentially localised protein in well-studied processes of early viral infection; such as, membrane trafficking and immune response.

We then carefully integrated several HCMV proteomic datasets and placed a spatial perspective on these data, including proteins targeted for degradation, as well as abundance and acetylation dataset. In addition, we augment a recent HCMV interactome by placing it in its spatial context and note that most host protein interactomes are in the same localisation as their viral bait. This provides an excellent resource for the community and highlights the benefit of integrating spatial proteomics and interactomics datasets.

Our analysis here highlights the potential role for post-translational modifications (PTMs) and their influence on localisation. The current datasets are limited because the spatial information is averaged over different PTMs. Thus, it is vital to develop methods to obtain spatial PTM information and develop corresponding computational tools to analyse these data. Furthermore, our approach here can only look at pairs of conditions at a time. In the future, more complex spatial proteomics designs will be available that will study multiple perturbations simultaneously and our approach will be adapted accordingly.

Overall, differential localisation experiments seek to add an orthogonal perspective to other assays, such as classical high-throughput differential abundance testing. Currently, differential localisation has not been extensively explored in high-throughput. We hope rigorous statistical methods will spur extensive and illuminating applications. An R-package is provided for analysis at https://ococrook.github.io/bandle/, building on a suite of packages for analysing spatial proteomics data[25,27,31].

## Methods

### The movement-reproducibility method

The movement-reproducibility (MR) method was proposed by refs. 17, 32 and this is our interpretation of their method. We suppose that we are given two spatial proteomics experiments under a single

contrast/perturbation/treatment, and denote unperturbed by $(d = 1)$ and $(d = 2)$ for the perturbed condition. Furthermore, assume we measure each condition with $r = 1, \ldots, R$ biological replicates. Let $X_1 = [X_1^{(1)}, \ldots, X_1^{(R)}]$ denote the concatenation of replicates for condition 1 and, likewise, for condition 2 we denote $X_2 = [X_2^{(1)}, \ldots, X_2^{(R)}]$. We first compute delta matrices as follows

$$\Delta = X_1 - X_2, \tag{1}$$

where $\Delta = [\Delta^{(1)}, \ldots, \Delta^{(R)}]$. This assumes that both features and replicates are comparable in some way; that is, a feature in the $r$th replicate is directly comparable to the same feature in another replicate. Then, for each $\Delta_r$, $r = 1, \ldots, R$, the squared Mahalanobis distance $D_M$ from each protein to the empirical mean is computed using a robust estimate of the covariance matrix—the minimum covariance determination method[123]. Under a Gaussian assumption on $\Delta_r$, $D_M(p_i)$ follows a chi-squared distribution with degrees of freedom equal to the dimension of the data $G$. Then, for each protein and each replicate a $p$-value is computed, such that there are $R$ such $p$-values for each protein. These $p$-values are combined into a score by taking the cube of the largest $p$-value for each protein, correcting for multiple hypothesis testing using the Benjamini–Hochberg procedure and computing the $-\log_{10}$ of the resultant value. For ref. 17, the $p$-value is not cubed and simply the largest $p$-value is taken. The final score is called the M score.

This process means that the computed value can no longer be interpreted as truly derived from a $p$-value. To maintain this interpretation, one could instead combine $p$-values using Fisher's method[45]. Furthermore, the authors are, implicitly, concerned with finding any false positives and as such control over the FWER is desired rather than the FDR. Since FWER $\geq$ FDR, control of the FDR does not lead to control over the FWER.

A so-called reproducibility (R) score is obtained by first computing the Pearson correlation pairwise between matrices $\Delta_i$, $\Delta_j$, $i \neq j$ for each protein. A final R score, for each protein, is obtained by taking the minimum value for each protein. Again this score could be interpreted in a formal testing procedure using a permutation test[124] and furthermore includes an assumption of bivariate normality. Moreover, Pearson's correlation is unresponsive to many non-linear relationships which might be present.

Finally, each protein has an associated pair of scores, referred to as the MR-score. To determine thresholds for these scores the authors take a desired FDR = 0.01. Thus they repeat a control experiment 6 times to determine thresholds $M = 2$, $R = 0.9$: a region with no false discoveries.

Repeating the control experiment 6 times is a costly process and likely to be prohibitive for most experiments, particularly for cells that are expensive to culture. Furthermore, since the thresholds are empirically derived, this process needs to be repeated for every new experiment to determine optimal thresholds.

## BANDLE
**A model for differential localisation.** In the following, we lay out our model for BANDLE, along with methods for inference, and approaches for summarising and visualising the output. Firstly, suppose we have two spatial proteomics experiments with unperturbed $(d = 1)$ and perturbed conditions $(d = 2)$. Furthermore, assume we measure each condition with $r = 1, \ldots, R$ biological replicates. Let $X_1 = [X_1^{(1)}, \ldots, X_1^{(R)}]$ denote the concatenation of replicates for condition 1 and likewise for condition 2 denotes $X_2 = [X_2^{(1)}, \ldots, X_2^{(R)}]$. We introduce the following latent allocation variable $z_{i,d}$, representing the localisation of protein $i$ in condition $d$. Thus, if $z_{i,d} = k$ this means that protein $i$ localises to organelle $k$ in dataset $d$. Given this latent allocation variable, we assume that the data from replicate $r = 1, \ldots, R$ arises from some component density $F(\cdot | \theta_k^{(r)})$. Hence, letting $\theta$ be the set of all component

parameters, we can write

$$x_{i,d}^{(r)} | z_{i,d}, \theta \sim F\left(x_{i,d}^{(r)} | \theta_{z_{i,d}}^{(r)}\right). \tag{2}$$

We assume that biological replicates are independent and so we factorise as follows

$$p(x_{i,d} | z_{i,d}, \theta) = \prod_{r=1}^{R} p(x_{i,d}^{(r)} | z_{i,d}, \theta_{z_{i,d}}^{(r)}). \tag{3}$$

To couple the two conditions together we assume a joint prior structure for the latent allocation variable in each dataset. To be more precise, we construct a prior for the pair $(z_{i,1}, z_{i,2})$. We fix the possible number of subcellular niches to which a protein may localise to be $K$. Now, we introduce the matrix Dirichlet distribution, which we denote as $\mathcal{M}\mathrm{Dir}(\alpha, K)$. The concentration parameter $\alpha$ is a $K \times K$ matrix, such that for a matrix $\boldsymbol{\pi}$, the pdf of the matrix Dirichlet distribution is

$$f(\boldsymbol{\pi} | \alpha) = \prod_{k=1}^{K} \frac{1}{\mathcal{B}(\alpha_k)} \prod_{j=1}^{K} \pi_{jk}^{\alpha_{jk} - 1}, \tag{4}$$

where $\mathcal{B}$ denotes the beta function, $\alpha_k$ denotes the $k$th row of $\alpha$, and $\sum_{j,k} \pi_{jk} = 1$. Thus, we propose the following hierarchical structure

$$\boldsymbol{\pi} | \alpha \sim \mathcal{M}\mathrm{Dir}(\alpha, K) \tag{5}$$

$$(z_{i,1}, z_{i,2}) \sim cat(\boldsymbol{\pi}), \tag{6}$$

where $(z_{i,1}, z_{i,2}) \sim cat(\boldsymbol{\pi})$ means that the prior allocation probabilities are given by

$$p(z_{i,1} = k, z_{i,2} = k' | \boldsymbol{\pi}) = \pi_{kk'}. \tag{7}$$

The above model is conjugate, and so if $n_{j,k} = |\{(z_{i,1}, z_{i,2}) = (j, k)\}|$, it follows that the conditional posterior of $\boldsymbol{\pi}$ is

$$\boldsymbol{\pi} | (Z_1, Z_2), \alpha \sim \mathcal{M}\mathrm{Dir}(\gamma, K), \tag{8}$$

where $\gamma_{j,k} = \alpha_{jk} + n_{j,k}$. The likelihood models for the data are Gaussian Random Fields, which we elaborate on in the following section. Hence, the conditional posterior of the allocation probabilities are given by

$$p(z_{i,1} = j, z_{i,2} = k | \boldsymbol{\pi}) \propto \pi_{jk} \prod_{r=1}^{R} p(x_{i,1}^{(r)} | z_{i,1} = j) p(x_{i,2}^{(r)} | z_{i,2} = k). \tag{9}$$

**Likelihood model.** The model described in the previous section is presented in a general form, so it could be applied to many different modes of data. We describe the model for a single spatial proteomics experiment, since the same model is assumed across all spatial proteomics experiments that are then subsequently joined together using the approach in the previous section. Though the model is the same across experiments, the parameters are experiment-specific.

We assume that the protein intensity $x_i$ at each fraction $s_j$ can be described by some regression model with unknown regression function:

$$x_i(s_j) = \mu_i(s_j) + \varepsilon_{ij}, \tag{10}$$

where $\mu_i$ is some unknown deterministic function of space and $\varepsilon_{ij}$ is a noise variable, which we assume is $\varepsilon_{ij} \sim \mathcal{N}(0, \sigma_i^2)$. Proteins are grouped together according to their subcellular localisation; such that, all proteins associated to subcellular niche $k = 1, \ldots, K$ share the same regression model. Hence, we write $\mu_i = \mu_k$ and $\sigma_i = \sigma_k$. Throughout, for clarity, we refer to subcellular structures, whether they are organelles,

vesicles or large protein complexes, as components. The regression functions $\mu_k$ are unknown and thus we place priors over these functions to represent our prior uncertainty. Protein intensities are spatially correlated and thus we place Gaussian random field (GRF) priors over these regression functions. We pedantically refer to these as GRF priors rather than Gaussian process (GP) priors to make the distinction between the 1D spatial process that separates subcellular niches and the experimental cellular perturbations, which are potentially temporal in nature. Hence, we write the following

$$\mu_k \sim GRF(m_k(\boldsymbol{s}), C_k(\boldsymbol{s}, \boldsymbol{s}')), \tag{11}$$

which is defined as:

**Definition 1**. *Gaussian Random Field*

*If $\mu(\boldsymbol{s}) \sim GRF(m_k(\boldsymbol{s}), C_k(\boldsymbol{s}, \boldsymbol{s}'))$ then for any finite dimensional collection of indices $s_1, \ldots, s_n$, $[\mu(s_1), \ldots, \mu(s_n)]$ is multivariate Gaussian with mean $[m(s_1), \ldots, m(s_n)]$ and covariance matrix such that $C_{ij} = C(s_i, s_j)$.*

Thus, each component is captured by a Gaussian Random Field model and the full complement of proteins as a finite mixture of GRF models. The protein intensity for each experiment might be measured in replicates. For a sufficiently flexible model, we allow different regression models across different replicates. To be more precise, consider the protein intensity $x_i^{(r)}$ for the $i$th protein measured in replicate $r$ at fraction $s_j^{(r)}$, then we can write the following

$$x_i^{(r)}\left(s_j^{(r)}\right) = \mu_k^{(r)}\left(s_j^{(r)}\right) + \varepsilon_{ij}^{(r)}, \tag{12}$$

having assumed that the $i$th protein is associated to the $k$th component. The (hyper)parameters for the Gaussian Random Field priors for the $r$th replicate in experiment $d$ are denoted by $\theta_{k,d}^{(r)}$. We denote by $\boldsymbol{\theta}$ the collection of all hyperparameters and the collection of priors for these hyperparameters by $G_0(\boldsymbol{\theta})$. The loss of conjugacy between the prior on the hyperparameters and likelihood is unavoidable.

The GRF is used to model the uncertainty in the underlying regression functions; however, we have yet to consider the uncertainty that a protein belongs to each of these components. To capture these uncertainties, we can use the model in the previous section, allowing information to be shared across each condition. Following from the previous section, the conditional posterior of the allocation probabilities is

$$p(z_{i,1} = j, z_{i,2} = k | \boldsymbol{\pi}) \propto \pi_{jk} \prod_{r=1}^{R} p\left(x_{i,1}^{(r)} | z_{i,1} = j\right) p\left(x_{i,2}^{(r)} | z_{i,2} = k\right), \tag{13}$$

where, in the specific case of our likelihood model the probabilities in the terms of the product can be computed using the appropriate GRF.

We assume that our GRFs are centred and that the covariance is from the Matern class[125]. The Matern covariance is specified as follows

$$C_\nu(d) = a^2 \frac{2^{1-\nu}}{\Gamma(\nu)} \left(\sqrt{8\nu}\frac{d}{\rho}\right)^\nu \mathcal{K}_\nu\left(\sqrt{8\nu}\frac{d}{\rho}\right), \tag{14}$$

where $\Gamma$ is the gamma function and $\mathcal{K}_\nu$ denotes the modified Bessel function of the second kind of order $\nu > 0$. Furthermore, $a$ and $\rho$ are positive parameters of the covariance. $a^2$ is interpreted as a marginal variance, while the non-standard choice of $\sqrt{8\nu}$ in the definition of the Matern covariance, allows us to interpret $\rho$ as a range parameter and thus $\rho$ is the distance at which the correlation is 0.1 for any $\nu$[126]. The Matern covariance arises from solutions of the following linear fractional stochastic partial differential equation (SPDE):

$$(\kappa^2 - \Delta)^{\alpha/2} x(u) = \mathcal{W}(u), \quad u \in \mathbb{R}^d \quad \alpha = \nu + d/2, \quad \kappa > 0, \quad \nu > 0, \tag{15}$$

where $\mathcal{W}(u)$ is spatial Gaussian white noise with unit variance and $\Delta$ is the Laplacian. The parameter $\nu$ controls the differentiability of the

resulting sample paths; such that, $\lceil \nu \rceil$ is the number of mean-square derivatives. For typical applications, $\nu$ is poorly identifiable and fixed; $\nu = 1/2$ recovers the exponential covariance, whereas taking the limit $\nu \to \infty$ one obtains the squared exponential (Gaussian) covariance. We fix $\nu = 2$.

A ridge in the marginal likelihood for the marginal variance and range parameters of the Matern covariance makes inference challenging. Indeed, different hyperparameters lead to unconditional prior simulations with the same spatial pattern but different scales[127,128]. Furthermore, when the intrinsic dimension of the Gaussian random field is less than four, there is no consistent estimator under in-fill asymptotics for $\rho$ and $a$. A principled prior, which allows domain expertise to be expressed, is thus desired to enable stable inferences. A number of works considered reference priors for GRFs[129–132]. Here, we employ a recently introduced collection of weakly-informative penalised complexity (PC) priors, which we explain in the next section.

**Penalised complexity priors for GRFs.** Here, we briefly described the PC priors used for the hyperparameters of the GRF models. Recall that $\nu$ models the smoothness and is fixed at 2. The idea behind the PC prior is to shrink the model towards a simpler model of lower complexity. In the case of GRF, these are models that cannot excessively curve, choosing to explain high frequency fluctuations with a wide variance. Fuglstad et al.[128] derive the appropriate PC prior as the following on the amplitude $a$ and the length-scale/range $\rho$:

$$\pi(a, \rho) = \frac{\lambda_1 \lambda_2}{2} \rho^{-3/2} \exp\left(-\lambda_1 \rho^{-1/2} - \lambda_2 a\right), \tag{16}$$

where $\lambda_1$ and $\lambda_2$ are hyperparameters that control shrinkage towards the simpler model. Further details are found in the Supplementary Methods. As a default, $\lambda_1 = 10$ and $\lambda_2 = 60$ and can be assessed by visual prior predictive checks[133]. The defaults were used throughout except for the simulated examples and the Ithzak dataset for which $\lambda_1 = 0.05$ was chosen by visual assessment.

**Penalised complexity prior for the noise model.** The noise effect is distributed according to $\epsilon_{ij} \sim \mathcal{N}(0, \sigma_k^2)$ for $k = 1, \ldots, K$. We additionally choose a PC prior in this scenario, first we reparametrize in terms of a precision $\tau_k = 1/\sigma_k^2$ for $k = 1, \ldots, K$. Then appealing to[134] the PC prior is a type-2 Gumbel distribution:

$$\pi(\tau) = \frac{\lambda_3}{2} \tau^{-3/2} \exp(-\lambda_3 \tau^{-1/2}). \tag{17}$$

The PC prior in this case shrinks towards zero variance. More details are found in the Supplementary Methods. As a default, $\lambda_3 = 250$ and can be assessed by visual prior predictive checks. For the Davies dataset $\lambda_3 = 200$, and for simulated examples and Itzhak dataset $\lambda_3 = 100$, which were chosen by visual assessment.

**Modelling outliers.** As shown in previous work, some proteins are not well described by any of the annotated components[30]. This could be because of undiscovered biological novelty, poor protein quantitation, the protein could reside in a yet to be described subcellular component or in multiple annotated compartments. To alleviate this issue we augment our model with an additional outlier component. We introduce a latent binary indicator $\phi_{i,d}$ to denote whether protein $i$ is better modelled as belonging to a known subcellular niche ($\phi_{i,d} = 1$) or a disperse outlier component ($\phi_{i,d} = 0$) in dataset $d$. Since an indicator can only take two values, it has a Bernoulli distribution and so we write $p_0(\phi_{i,d} = 0) = \varepsilon_d$. As in previous work, we model the density of the outlier component by a student's-t distribution with degrees of freedom 4, mean equal to the empirical mean and the covariance to be the empirical covariance of the data. We also assume the covariance

matrix to be diagonal. Finally, we place a Beta prior on $\epsilon_d \sim B(u_d, v_d)$ allowing us to specify a prior number of outliers. We opt for a previously recommended weakly-informative prior $u_d = 2$ and $v_d = 10$ for $d = 1, 2$[30]. The relevant conditional distributions for MCMC sampling are found in the Supplementary Methods.

**Bandle in hierarchical model notation.** To summarise the specification of the model, we display the bandle model in the following Bayesian Hierarchical model:

$$x_{i,d}^{(r)}|z_{i,d} = k, \theta, \phi_{i,d} = 1 \sim \mu_k^{(r)}(s_j) + \epsilon_{kj}^{(r)}$$
$$x_{i,d}^{(r)}|z_{i,d} = k, \theta, \phi_{i,d} = 0 \sim \mathcal{T}(4, M, V)$$
$$\mu_{k,d}^{(r)} \sim GRF(m_{k,d}^{(r)}(s), C_{k,d}^{(r)}(s, s'))$$
$$\epsilon_{kj}^{(r)} \sim \mathcal{N}(0, \sigma_{r,k}^2)$$
$$1/\sigma_{r,k}^2 \sim \text{Type-2 Gumbel}(\lambda_3)$$
$$(z_{i,1}, z_{i,2}) \sim cat(\pi) \qquad (18)$$
$$\pi|\alpha \sim \mathcal{M}Dir(\alpha, K)$$
$$\phi_{i,d} \sim Ber(\epsilon_d)$$
$$\epsilon_d \sim B(u_d, v_d)$$
$$C_v(\delta) = a^2 \frac{2^{1-\nu}}{\Gamma(\nu)} \left(\sqrt{8\nu} \frac{\delta}{\rho}\right)^\nu \mathcal{K}_v \left(\sqrt{8\nu} \frac{\delta}{\rho}\right)$$
$$a_{k,d}^{(d)}, \rho_{k,d}^{(d)} \sim PC(\lambda_1, \lambda_2).$$

In the above equation $\mathcal{T}$ denotes the student-t distribution, $Ber$ denotes the Bernoulli distribution, $B$ the Beta distribution, $\mathcal{K}_v$ the Bessel function of the 2nd kind with parameter $v$ and $PC$ denotes the penalised complexity prior for a GRF. The other notation is as described in the previous section. For details of the Pólya-Gamma-based model, we refer the Supplementary Methods and Supplementary Note 12.

**Major algorithmic steps of bandle.** The complex notation in the previous section can be cumbersome and so here, we summarise the major steps of bandle in algorithmic steps:

1. First, for each subcellular niche $k$ in each dataset $d$ and replicate $r$, learn the GRFs and corresponding hyperparameters $a$ and $\rho$ by maximum a posteriori estimation for a pre-specified $\lambda_1, \lambda_2$, as well as the variance parameter $\sigma^2$ for pre-specified $\lambda_3$. $\lambda_1, \lambda_2$ and $\lambda_3$ can be selected using prior predictive checks or default choices.
2. Select $\alpha$, potentially using a prior predictive check, expert knowledge or default choices.
3. For $T$ Monte-Carlo iterations perform the following steps ((a)–(i)). Note that in each case the currently sampled parameters are used in the following step and probabilities are conditioned on these sampled parameters. The dependence on $T$ and these sampled parameters is suppressed to avoid notational clutter.
(a) Compute the likelihood of each protein $i$ belonging to subcellular niche $k$ in replicate $r$ for dataset $d$. That is for $k = 1, \ldots, K$, $i = 1, \ldots, n$, $d = 1, 2$ and $r = 1, \ldots, R$, compute

$$p(x_{i,d}|z_{i,d} = k, \theta) = \prod_{r=1}^R p(x_{i,d}^{(r)}|z_{i,d} = k, \theta_{z_{i,d}}^{(r)}). \qquad (19)$$

(b) Sample from the conditional posterior distribution of $\pi$. Letting $Z$ be the set of allocation of all proteins to all niches and $\alpha'$ be the matrix which tabulates the total allocation to each pair of niches, then:

$$\pi'|\alpha, Z \sim \mathcal{M}Dir(\alpha + \alpha', K). \qquad (20)$$

(c) Compute the conditional posterior of each protein $i$ belonging to each subcellular niche $k$ in each dataset $d = 1, 2$, using the following equation:

$$p(z_{i,1} = j, z_{i,2} = k|\pi', \theta, x_{i,d}) \propto \pi'_{jk} \prod_{r=1}^R p(x_{i,1}^{(r)}|z_{i,1} = j, \theta)p(x_{i,2}^{(r)}|z_{i,2} = k, \theta). \qquad (21)$$

(d) Sample $(z_{i,1}, z_{i,2})$ from a categorical distribution using the above computed conditional posterior probabilities.
(e) Sample $\epsilon_d$ from the conditional posterior distribution. The formula is given by

$$\epsilon_d' \sim \mathcal{B}(u_d + u_d', v_d + v_d'), \qquad (22)$$

where $u_d'$ is the current number of proteins allocated to the outlier component in dataset $d$ and $v_d'$ is the current number of proteins not allocated to the outlier component in dataset $d$.
(f) Compute the conditional posterior of belonging to the outlier component, using the following equation:

$$p(\phi_{i,d}|z_{i,1} = k, z_{i,2} = j, \theta, x_{i,d}, \epsilon_d') \propto (1 - \epsilon_d') \prod_{r=1}^R F(x_{i,d}^{(r)}|\theta_k^{(r)})$$
$$+ \epsilon_d' \prod_{r=1}^R G(x_{i,d}^{(r)}|\Phi^{(r)}), \qquad (23)$$

where $\Phi^{(r)}$ denotes the parameters of the outlier component for replicates $r$, while $F$ and $G$ are the densities of the niche-specific component and outlier component, respectively.
(g) Sample $\phi_{i,d}$ from a Bernoulli distribution for all $i$ and all $d$ from the conditional posterior probabilities given by the above equation.
(h) Optional: Sample new GRF hyperparameters from the conditional posterior distribution using Metropolis-Hastings or Hamiltonian Monte-Carlo. Otherwise use precomputed values. This conditional posterior is given by, for each dataset $d$, replicate $r$ and niche $k$:

$$p\left(a_{r,k}^{(d)}, \rho_{r,k}^{(d)}, \sigma_{r,k}^2|X, \lambda\right) \propto p_0\left(a_{r,k}^{(d)}, \rho_{r,k}^{(d)}, \sigma_{r,k}^2|\lambda\right) p\left(X|a_{r,k}^{(d)}, \rho_{r,k}^{(d)}, \sigma_{r,k}^2\right) \qquad (24)$$

Note that $p(X|a_{r,k}^{(d)}, \rho_{r,k}^{(d)}, \sigma_{r,k}^2)$ is given by a Gaussian density, with equation given in the supplement. The prior on the GRF hyperparameters is given by the penalised complexity priors. The lack of conjugacy between the prior and likelihood means a Metropolis-Hastings or Hamiltonian Monte-Carlo move is required.
(i) Update the GRF distributions for each subcellular niche $k$ in replicate $r$ for dataset $d$. That is, sample $\mu_{k,d}^{(r)}$ from the conditional posterior GRFs. Equations are given in Rasmussen and Williams[127] and ref. [36].

**Calibration of Dirichlet prior.** The following section describes how to calibrate the prior based on expert information and prior predictive checks. Recall the prior on the allocation probabilities is the following

$$p(z_{i,1} = k, z_{i,2} = k'|\pi) = \pi_{kk'}. \qquad (25)$$

The matrix $\pi$ has $\pi_{jk}$ has its $(j, k)^{th}$ entry and $\pi_{jk}$ is the prior probability that a protein belongs to organelle $j$ in dataset 1 (control) and $k$ in dataset 2 (contrast). The diagonal terms represent the probability that the protein was allocated to the same organelle in each dataset. The non-diagonal terms are the prior probability that the protein was not allocated to the same organelle. Since the number of non-diagonal terms greatly exceeds the number of diagonal entries, it is important

to specify this prior carefully. Recall that the prior is given a matrix Dirichlet distribution with concentration parameter $\alpha$.

Firstly, we are interested in the prior expectation of the number of proteins that are differentially localised; that is, proteins not allocated to the same organelle in both conditions. Let $\gamma$ be the prior probability that a protein is not allocated to the same organelle. Then it follows that

$$p(z_{i,1} \neq z_{i,2} | \boldsymbol{\pi}) =: \gamma = \sum_{j,k;j\neq k} \pi_{jk}. \qquad (26)$$

By properties of the Dirichlet distribution, we have that the marginal distribution of $\pi_{jk}$ is given by

$$\pi_{jk} \sim \mathcal{B}(\alpha_{jk}, \alpha_0 - \alpha_{jk}), \qquad (27)$$

where $\alpha_0 = \sum_{j,k} \alpha_{jk}$. Thus, the expected value of $\gamma$ is computed as follows

$$\begin{aligned} \mathbb{E}[\gamma] &= \sum_{j,k;j\neq k} \mathbb{E}[\pi_{jk}] \\ &= \sum_{j,k;j\neq k} \frac{\alpha_{jk}}{\alpha_0}. \end{aligned} \qquad (28)$$

We are further interested in the probability that a certain number of proteins, say $q$, are differential localised. Letting $N_U$ be the number of unlabelled proteins in the experiment, then the distribution of the prior number of differential localised proteins is

$$p(N_U \gamma > q) = p\left(N_U \sum_{j,k;j\neq k} \pi_{jk} > q\right) = \delta. \qquad (29)$$

Computing $\delta$ is not simple; however, it is straightforward to estimate $\delta$ using Monte-Carlo, by simply sampling from Beta distributions:

$$p\left(N_U \sum_{j,k;j\neq k} \pi_{jk} > q\right) \approx \frac{1}{T}\sum_{t=1}^{T} \mathbb{1}\left(N_U \sum_{j,k;j\neq k} \pi_{jk}^{(t)} > q\right). \qquad (30)$$

Thus, we calibrate the Dirichlet prior using the above expectation and quantile. In some applications, calibrating several quantiles is needed to ensure sufficient mass is placed on desired regions of the probability space. For example, let $q_1 < q_2$, then we want that $\delta_1$, below, is not so small to rule out reasonable inferences and that $\delta_2 < \delta_1$ is sufficiently large. These can be computed from the equations below:

$$p\left(N_U \sum_{j,k;j\neq k} \pi_{jk} > q\right) \approx \frac{1}{T}\sum_{t=1}^{T} \mathbb{1}\left(N_U \sum_{j,k;j\neq k} \pi_{jk}^{(t)} > q_1\right) = \delta_1, \qquad (31)$$

$$p\left(N_U \sum_{j,k;j\neq k} \pi_{jk} > q\right) \approx \frac{1}{T}\sum_{t=1}^{T} \mathbb{1}\left(N_U \sum_{j,k;j\neq k} \pi_{jk}^{(t)} > q_2\right) = \delta_2. \qquad (32)$$

More precise and informative prior biological knowledge can be specified; for example, should we suspect that some relocalisation events between particular organelles are more likely than others due to the stimuli, these can be encoded into the prior. If we expect more relocalisation events between organelle $j$ and $k_1$ than organelle $j$ and $k_2$, this can be encoded by ensuring

$$\frac{1}{T}\sum_{t=1}^{T} \mathbb{1}\left(\pi_{jk_1}^{(t)} > \pi_{jk_2}^{(t)}\right) > \delta_3 > 0. \qquad (33)$$

For example, suppose that for a particular experiment it is impossible for any protein to relocalise. Then, we are interested in

ensuring

$$p\left(N_U \sum_{k,j;j\neq k} \pi_{jk} > 0\right) = \delta = 0. \qquad (34)$$

This is only possible if $\pi_{jk} = 0$ for all $k \neq j$. This can be ensured by setting $\alpha_{jj} = 1$ and $\alpha_{jk} = 0$ for $k \neq j$. Now suppose, we wish to relax this assumption slightly, since we believe some re-localisations are possible. In our experiment, we measure 1000 unlabelled proteins and we believe that there is roughly a 10% chance there are more than 10 re-localisations. Then we wish to satisfy

$$p\left(1000 \sum_{k,j;j\neq k} \pi_{jk} > 10\right) = 0.1. \qquad (35)$$

The appropriate entries of $\alpha$ can then be chosen using Monte-Carlo simulation. Alternatively, if an objective Bayesian analysis is preferred, the Jeffery's prior sets $\alpha_{jk} = 0.5$ for every $j, k = 1, \ldots, K$. This approach is not generally recommended by the authors, because the diagonal terms of $\boldsymbol{\pi}$ have a different interpretation to the off-diagonal terms. As a default, we set $\alpha_{jj} = 1.01$ and $\alpha_{jk} = 0.01$ for $k \neq j$. This assumes that there are roughly an order of magnitude fewer differentially localised proteins than spatially stable ones. This default was used in all simulations and applications except the EGF simulation dataset. In that case, we had prior knowledge of a differentially localisation between the plasma membrane and the endosome and so we set the corresponding entry of $\alpha$ to 1. Additional details of setting of the priors for specific application is in the Supplementary Material (Supplementary Note 11). We also demonstrate that our analysis remains robust to the choice of prior by performing a prior sensitivity analysis (Supplementary Note 11). Convergence analysis of MCMC algorithms is provided in Supplementary Notes 6, 8 and 10.

**Differential localisation probability.** The main posterior quantity of interest is the probability that a protein is differentially localised. This can be approximated from the $T$ Monte-Carlo samples as follows, suppressing notational dependence on all data and parameters for clarity

$$\chi_i = p(z_{i,1} \neq z_{i,2}) \approx \frac{1}{T}\sum_{t=1}^{T} \mathbb{1}\left(z_{i,1}^{(t)} \neq z_{i,2}^{(t)}\right), \qquad (36)$$

where $t$ denotes the $t$th sample of the MCMC algorithm. It is important to note that this quantity is agnostic to the assigned subcellular niche. We notice that the distribution of the number of MCMC observations for which $z_1$ is not equal to $z_2$ is given by:

$$\sum_{t=1}^{T} \mathbb{1}\left(z_{i,1}^{(t)} \neq z_{i,2}^{(t)}\right) \sim \mathcal{B}(\chi_i, T). \qquad (37)$$

Hence, in this case, the Monte-Carlo estimator for $\chi$ is simply the maximum likelihood estimator of the probability parameter of the above binomial distribution. As $T$ is given, uncertainty estimates, such as credible intervals, can be obtained from this binomial distribution directly.

An alternative, but less computationally efficient approach, to perform uncertainty quantification on the differential localisation probability, could use the non-parametric bootstrap on the Monte-Carlo samples. More precisely, first sample uniformly with replacement from $\{z_{i,1}^{(t)}\}_{t=1}^{T}$ and $\{z_{i,2}^{(t)}\}_{t=1}^{T}$ to total of $T$ samples. This produces a bootstrap sample indexed by $B_1$. Then compute our statistic

of interest:

$$\chi^{*}_{i,B_1} \approx \frac{1}{|B_1|} \sum_{t \in B_1} \mathbb{1}(z^{(t)}_{i,1} \neq z^{(t)}_{i,2}). \quad (38)$$

This process is then repeated to obtain a set of bootstrap samples $\mathbb{B} = \{B_1,\ldots,B_b\}$, for some large $b$, say 1000. For each $B_r \in \mathbb{B}$, we compute $\chi^{*}_{i,B_r}$ for $r = 1,\ldots,b$, obtaining a sampling distribution for $\chi_r$ from which we can compute functionals of interest.

**Posterior localisation probabilities.** A further quantity of interest is the posterior probability that a protein belongs to each of the $K$ subcellular niches present in the data. For the control, this is given by the following Monte-Carlo average

$$p(z_{i,1} = k|\Theta) \approx \frac{1}{T} \sum_{t=1}^{T} p\left(z^{(t)}_{i,1} = k|\Theta\right), \quad (39)$$

where $\Theta$ denotes all other quantities in the model. A corresponding formula also holds for the second dataset

$$p(z_{i,2} = k|\Theta) \approx \frac{1}{T} \sum_{t=1}^{T} p\left(z^{(t)}_{i,2} = k|\Theta\right). \quad (40)$$

The posterior distribution of these quantities and uncertainty estimates can be computed and visualised in standard ways.

**The BANDLE package.** The BANDLE package (https://bioconductor.org/packages/release/bioc/html/bandle.html) is implemented as part of the Bioconductor suite[135], with documentation covering a typical analysis workflow, including the analysis of the spatio-temporal proteome of THP-1[136]. The package includes utility functions to set priors, as well as data visualisations of the outputs of BANDLE. BANDLE is designed to accompany the MSnbase, pRoloc, pRolocGUI and pRolocdata packages[25,27], as part of an integrated Bioconductor suite. Our pipeline offers a modular and extensible approach that includes all steps of the spatial proteomics workflow. This includes aggregation of peptide-spectrum matches (PSMs) to proteins, assessment of data quality[40], imputation, normalisation, unsupervised analysis, clustering, supervised machine learning, transfer learning[24], semi-supervised learning[22,23,36], differential localisation analysis, data management and dissemination, as well as analysis-specific visualisation. This framework also allows for seamless deployment into Shiny applications as metadata is stored in a consistent manner, for example see https://proteome.shinyapps.io/thp-lopit/. The package also provides an implementation of the MR method.

**Mass-spectrometry data processing**

For the EGF stimulated HeLa cells data found in section: Characterising differential localisation upon EGF stimulation, quantitation was performed using SILAC. Protein level intensities were exported from MaxQuant. Proteins with any missing values were removed from the analysis. We refer to Itzhak et al.[17] for further information. For the AP-4 knockout dataset in section: BANDLE obtains deeper insights into AP-4 dependent localisation, quantitation was performed using SILAC and protein level intensities were exported from MaxQuant. We refer to Davies et al.[4] for more information. For the HCMV application in section: Rewiring the proteome under cytomegalovirus infection, quantitation was performed using TMT and protein level intensities were exported from MaxQuant for further details see Beltran et al.[16].

**Reporting summary**

Further information on research design is available in the Nature Research Reporting Summary linked to this article.

## Data availability

The spatial proteomics data have been deposited and is available in the Bioconductor package pRolocdata. The additional data are given in the referenced manuscripts and additionally are provided as part of the Supplementary Material. The MCMC data generated in this study have been deposited in Zenodo: https://doi.org/10.5281/zenodo.4415369, supplementary code and data are deposited at Zenodo: https://doi.org/10.5281/zenodo.6514300. PXD010103 (The AP-4 dataset). PXD003925 (The HCMV spatial protemics dataset). PXD009839 (The HCMV acetylation dataset). PXD009945 (The HCMV degradation assays datasets). PXD014845 (The HCMV interactome dataset). String version 11.5 database was used to retrieve annotations and pathway enrichment results (https://string-db.org/). Source data are provided with this paper.

## Code availability

An R-package is provided at https://bioconductor.org/packages/release/bioc/html/bandle.html. R version 4.1 was used in the analysis of the data. BANDLE is archived at https://zenodo.org/badge/latestdoi/324635075[137].

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

## Acknowledgements

O.M.C. is a Wellcome Trust Mathematical Genomics and Medicine student funded by the Cambridge School of Clinical Medicine. As well as MRC awards to P.D.W.K. MC_UU_00002/13. K.S.L. is supported by the Wellcome Trust 110071/Z/15/Z. The funders had no role in study design, data collection and analysis, decision to publish, or preparation of the manuscript. This work was supported by the National Institute for Health Research [Cambridge Biomedical Research Centre at the Cambridge University Hospitals NHS Foundation Trust] [*]. *The views expressed are those of the authors and not necessarily those of the NHS, the NIHR or the Department of Health and Social Care.

## Author contributions

O.M.C. developed the statistical methodology with advice from P.D.W.K. O.M.C. and L.M.B. developed the software with guidance from L.G. O.M.C. wrote the manuscript with guidance from K.S.L. O.M.C. and C.D. interpreted the data and results. O.M.C. conceived the project and analysed the data. J.C. provided feedback on the software. All authors edited the manuscript and shaped the narrative. K.S.L., L.G. and P.D.W.K. supervised the project.

## Competing interests

Colin T.R. Davies is an employee of AstraZenca (AZ). AZ had no role in the study design, data collection, analysis, decision to publish, or preparation of the manuscript. All other authors declare no competing interests.
