## [Peer Review File · Nature Communications]

REVIEWER COMMENTS

Reviewer #3 (Remarks to the Author):

I was asked specifically to check if the concerns raised by Reviewers #1 and #2 were sufficiently addressed in the updated manuscript. In general, the work looks solid and provides an elegant way of modeling the data. However, the concerns raised by the reviewers with regard to the presentation of the method and the validation of the estimates still require additional work.

The concerns raised by Reviewer #1 regarding the quality of the software and computational reproducibility have been adequately addressed by the integration of BUNDLE into Bioconductor and the deposition into the Zenodo repositories. The presentation of the method in the main manuscript is generally possible to follow but still requires some improvements:

- In section 4.2.3, explanations of some variables are still missing. For example, the outlier component and its corresponding variables (ϕ , e_d , u_d , v_d , M , V) are not mentioned at all in section 4.2 nor referred to as being found in the supplement. Some (short) description of this part of the model belongs in the main text anyway. The same is true for the lambda hyperparameters and it's not clear where or why the kappa variable comes into the model. There should also be a reference for the penalized complexity prior in the main text and perhaps a short description on the intuition behind it.
- In section 4.2.4, one can be much more explicit in which variables or conditional distributions are meant by using mathematical notation. Are the GRF hyperparameters that are learned kappa and rho? I'm also confused about the meaning of "pre-specified" lambdas. Are they fixed to a specific value at this stage? If so, how are they chosen?
- In section 4.2.5, it's not clear how the different deltas should be set/used by the user in practice. An example might be helpful here.

The issues raised by Reviewer #2 regarding the influence of the prior and validation of the probability estimates have also not been fully addressed.

- In the case of EGF stimulation where an informed prior was used, it would be beneficial to at least show the results of e.g. using the Dirichlet prior in the supplement to ensure that the results are not completely driven by the prior. The fact that an informed prior was used in this case should also explicitly be mentioned in the results section already. Generally, it's still unclear which priors were used for which results. It's easy to get lost in the supplementary methods, not knowing which techniques were actually used, which are just suggestions and under what circumstances one can/should use which.
- The expected calibration error does not seem to be the best metric to show accuracy of the probabilities. For example, it could simply be dominated by the non-translocated predictions, i.e. what is the ECE if one just predicts zero translocation probability for all proteins? If one sticks with the ECE, it could be clarified by plots showing the calibration error for the different bins. However, as Reviewer #2 already seems to allude to, one can also sort the proteins by localization posterior error probability (PEP) and compute the FDR as the average of the PEPs (<https://doi.org/10.1021/pr700739d>). This FDR can then be compared to the ground truth in the simulation studies and should ideally be shown to produce conservative estimates. I acknowledge that this might be a too high standard to ask for and the number of translocations is too small to produce smooth FDR estimates. However, such an analysis would at least give users a rough idea how much they can trust the probability estimates.

Minor comments:

- In section 4.2.5, the re-use of the rho variable is potentially confusing with the hyperparameter rho in the previous section.
- The term “feature” appears several times without explicitly explaining what they refer to.
- When introducing Gaussian Random Fields on page 4, it would be helpful to already mention its connection to Gaussian Processes which is a much more widely known concept. Without being too familiar with the exact terminology, I also don't understand the need for calling it a Gaussian Random Field, even if one is being “pedantic” as the authors put it. It seems that the time dimension is never modeled with a Gaussian Process, so calling it a Gaussian Random Field made it more confusing for me, as I was searching for that extra dimension.
- Figure 1d: can a differential localization probability of ~ 0.4 really be considered stable?
- Figure 3a: it is unclear which translocations we should expect as all translocated proteins have the same marker.
- Figure 5: mix of uniprot IDs and gene names is confusing
- The authors mention several times that they only looked at two conditions for the sake of brevity, but can BUNDLE actually handle more than two conditions? To me, it's not completely trivial how one would choose and evaluate priors in this case.

Software comments:

- In the documentation of fitGP, there is no explanation of how to choose a prior for the hyppar parameter (prior, length-scale amplitude and standard deviation). It shouldn't be necessary to look into the Crook et al. 2021 publication.
- For the authors' information, I had to install libxml2-dev, zlib1g-dev, libnetcdf-dev to install BUNDLE in an R Docker instance, perhaps this can be documented somewhere as the R error and warning messages are not the most helpful.
- In vignette 1, section 6.5, the authors set an 1% FDR threshold on the differentially localized proteins, but then simply use a cutoff of 0.99. In reality, the FDR is the average error probability of the ranked proteins.

Spelling/Grammar:

- Page 18 endsomes -> endosomes
- Page 25 and those that are not -> than those that are not
- Page 26 cell states -> cell state; those were -> those where ... was
- Page 28 dependant -> dependent?
- Page 28 is obtain -> is obtained; be interpret -> been interpreted

To the Editor and Reviewers of

Inferring differential subcellular localisation in comparative spatial proteomics using BUNDLE

We are grateful for the time and expertise of the reviewers who have thoroughly considered our manuscript. The reviewers' feedback raised a number of valuable points, which can be summarised as follows:

- 1) Complete mathematical descriptions of the model
- 2) Sensitivity to the choice of prior
- 3) Validation of probability estimates
- 4) Suggestions for figure, textual and software package clarity.

We believe that all these points were incredibly valuable to the revision of our manuscript and hence we have attempted to fully incorporate the reviewers' feedback into the re-submitted version. At a high level, this includes a number of changes. Firstly, we have edited the manuscript to cover the mathematical descriptions in more detail and move more content to the main text for clarity. We have performed additional analysis to highlight and better display the sensitivity of the method dependence on the choice of prior. We have included an analysis of false discovery proportion as estimated from the differential localisation probabilities. We have made several edits to the figures, text and software to improve clarity and useability. We believe that our manuscript is considerably improved and is appropriate to the wide readership of *Nature Communications*.

We attach a point-by-point response to the reviewer's suggestions. Our comments specifically to the reviewers are written in blue and the edited text in red. A marked up copy of our manuscript is also attached with new text marked in red.

**Yours faithfully,
Dr Oliver Crook**

Reviewer #3 (Remarks to the Author):

I was asked specifically to check if the concerns raised by Reviewers #1 and #2 were sufficiently addressed in the updated manuscript. In general, the work looks solid and provides an elegant way of modeling the data. However, the concerns raised by the reviewers with regard to the presentation of the method and the validation of the estimates still require additional work.

We thank the reviewers for their positive evaluation of our work and we are particularly grateful to them for clarifying some of the other reviewers' comments. We are glad the reviewer found our work elegant and we apologize for there being remaining concerns. The reviewer will find below a point-by-point response to their feedback in which we have explained how we have incorporated the feedback into a revised manuscript. We feel that the manuscript has significantly improved as a result and we hope the manuscript is now deemed suitable for publication.

The concerns raised by Reviewer #1 regarding the quality of the software and computational reproducibility have been adequately addressed by the integration of BUNDLE into Bioconductor and the deposition into the Zenodo repositories. The presentation of the method in the main manuscript is generally possible to follow but still requires some improvements:

- In section 4.2.3, explanations of some variables are still missing. For example, the outlier component and its corresponding variables (ϕ , e_d , u_d , v_d , M , V) are not mentioned at all in section 4.2 nor referred to as being found in the supplement. Some (short) description of this part of the model belongs in the main text anyway. The same is true for the λ hyperparameters and it's not clear where or why the κ variable comes into the model. There should also be a reference for the penalized complexity prior in the main text and perhaps a short description on the intuition behind it.

The reviewer raises an important point and we apologize for omitting a description of the outlier component in the main text. We have added the following text to the main text:

As shown in previous work, some proteins are not well described by any of the annotated components \cite{Crook:2018}. This could be because of undiscovered biological novelty, poor protein quantitation, the protein could reside in a yet to be described sub-cellular component or in multiple annotated compartments. To alleviate this issue we augment our model with an additional outlier component. We introduce a latent variable indicator $\phi_{i,d}$ to denote whether protein i is better modelled by a subcellular niche or a disperse outlier component in dataset d . Since an indicator can only take two values, it has a Bernoulli distribution and so we write $p_0(\phi_{i,d} = 0) = \epsilon_d$. As in previous work, we let the density of the outlier component be modelled by a student's t-distribution with degrees of freedom ν , mean equal to the empirical mean and the covariance to be the empirical covariance of the data. We also assume the covariance matrix to be diagonal. Finally, we place a Beta prior on $\epsilon_d \sim B(u_d, v_d)$ allowing us to specify a prior number of outliers. We opt for a previously recommended weakly-informative prior $u_d = 2$ and $v_d = 10$ for $d = 1, 2$

\cite{Crook:2018}. The relevant conditional distributions for MCMC sampling are found in the supplementary methods.

The lambda parameters were also relegated to the supplementary material but now have some explanation in the main text along with a description of the penalized complexity prior. We have added the following text to the methods:

Here, we briefly described the PC priors used for the hyperparameters of the GRF models. Recall that ν models the smoothness and is fixed at 2. The idea behind the PC prior is to shrink the model towards a simpler model of lower complexity. In the case of GRF, these are models that cannot excessively curve, choosing to explain high frequency fluctuations with a wide variance. \cite{Fuglstad::2019} derive the appropriate PC prior as the following on the amplitude a and the length-scale/range ρ :

$$\begin{equation} \pi(a, \rho) = \frac{\lambda_1 \lambda_2}{\rho^3} \exp(-\lambda_1 \rho^{-1/2} - \lambda_2 a), \end{equation}$$

where λ_1 and λ_2 are hyperparameters that control shrinkage towards the simpler model. Further details are found in the supplementary methods. As a default, $\lambda_1 = 10$ and $\lambda_2 = 60$ and can be assessed by visual prior predictive checks \cite{Gabry::2017}. The defaults were used throughout except for the simulated examples and the lthzak dataset for which $\lambda_1 = 0.05$ was chosen by visual assessment.

And

The noise effect is distributed according to $\epsilon_{ij} \sim \mathcal{N}(0, \sigma_k^2)$ for $k=1, \dots, K$. We additionally choose a PC prior in this scenario, first we reparametrize in terms of a precision $\tau_k = 1/\sigma_k^2$ for $k = 1, \dots, K$. Then appealing to \cite{Simpson::2017} the PC prior is a type-2 Gumbel distribution:

$$\begin{equation} \pi(\tau) = \frac{\lambda_3}{\tau^3} \exp(-\lambda_3 \tau^{-1/2}). \end{equation}$$

The PC prior in this case shrinks towards zero variance. More details are found in the supplementary methods. As a default, $\lambda_3 = 250$ and can be assessed by visual prior predictive checks. For the Davies dataset $\lambda_3 = 200$, and for simulated examples and lthzak dataset $\lambda_3 = 100$, which were chosen by visual assessment.

Kappa appears from reparameterization of the model to make computations more amenable but it is a mistake to introduce this in the main text, as it is simply mathematical convenience rather than reformulation that helps with interpretation. The formula can be found in the supplementary

material, but reference to it in the main text has been removed. Thank you for spotting this inconsistency.

- In section 4.2.4, one can be much more explicit in which variables or conditional distributions are meant by using mathematical notation. Are the GRF hyperparameters that are learned κ and ρ ? I'm also confused about the meaning of "pre-specified" lambdas. Are they fixed to a specific value at this stage? If so, how are they chosen?

This is another good point raised by the reviewer. Whilst a and ρ are the hyperparameters of the GRF, the lambdas are the parameters of the hyperprior on a and ρ . This has been clarified by the text in the response to the previous comment. There are default choices for the lambdas but GRFs are also amenable to visualize prior predictive checks for which there is support in the BUNDLE package. We have added the following text:

First, for each subcellular niche k in each dataset d and replicate r , learn the GRFs and corresponding hyperparameters a and ρ by `\textit{maximum a posteriori}` estimation for a pre-specified λ_1, λ_2 , as well as the variance parameter σ^2 for pre-specified λ_3 . λ_1, λ_2 and λ_3 can be selected using prior predictive checks or default choices.

Whilst we initially preferred section 4.2.4 (4.2.7) without extensive mathematical notation, we have now added the various conditional distributions to that section.

- In section 4.2.5, it's not clear how the different δ s should be set/used by the user in practice. An example might be helpful here.

We have added an example to the text and other examples are included in the vignettes of the package:

For example, suppose that for a particular experiment it is impossible for any protein to relocalise. Then, we are interested in ensuring

$$\begin{equation} \mathbb{P}\left(\bigcup_{k,j; j \neq k} \pi_{jk} > 0\right) = \delta = 0. \end{equation}$$

This is only possible if $\pi_{jk} = 0$ for all $k \neq j$. This can be ensured by setting $\alpha_{jj} = 1$ and $\alpha_{jk} = 0$ for $k \neq j$. Now suppose, we wish to relax this assumption slightly, since we believe some re-localisations are possible. In our experiment, we measure 1000 unlabelled proteins and we believe that there is roughly a 10% chance there are more than 10 re-localisations. Then we wish to satisfy

$$\begin{equation} \mathbb{P}\left(1000 \sum_{k,j; j \neq k} \pi_{jk} > 10\right) = 0.1. \end{equation}$$

The appropriate entries of α can then be chosen using Monte-Carlo simulation.

The issues raised by Reviewer #2 regarding the influence of the prior and validation of the probability estimates have also not been fully addressed.

- In the case of EGF stimulation where an informed prior was used, it would be beneficial to at least show the results of e.g. using the Dirichlet prior in the supplement to ensure that the results are not completely driven by the prior. The fact that an informed prior was used in this case should also explicitly be mentioned in the results section already. Generally, it's still unclear which priors were used for which results. It's easy to get lost in the supplementary methods, not knowing which techniques were actually used, which are just suggestions and under what circumstances one can/should use which.

This is a valuable point raised by the reviewer. We have reperformed the analysis on the EGF stimulation using the default Dirichlet prior. We find that the known translocations are found by both methods (EGFR, SHC-1 and GRB2). Proteins that had probability > 0.99 of differential localisation in the informative prior scenario also had probability > 0.99 of differential localisation in the default scenario. This thus confirms that new differential localisations were not being driven by the informative prior. In general, we found a very high correlation between the differential localisation probabilities between both scenarios (0.9, Spearman's correlation). We have added figures to the supplementary material and the following text:

Prior sensitivity for EGF stimulation case study:

To confirm that the results in the EGF stimulation case study was not completely driven by the use of an informative prior, we reperformed the analysis using the default prior choice. Whilst the differential localisations probabilities are unsurprisingly different, the major conclusions are robust to the choice of prior. We found that the known differential localisations are found by BUNDLE with either choice of prior with probability > 0.99 (EGFR, SHC-1 and GRB2). To confirm that additional differential localisations are not being stated due to the informative prior, we found that all proteins with probability greater than 0.99 of being differentially localized in the informative prior case were also found to have probability greater than 0.99 when the default prior was used. We found a high correlation (0.90, Spearman's correlation) between the differential localisation probabilities in the default prior and informative prior situation. We performed principal component regression and determined that the differential localisation probabilities tend to be slightly higher when using the default prior (see figure).

Figure caption: Comparing priors for EGF application

(left) Boxplots of differential localisation probabilities when using the default Dirichlet prior.

(center) Boxplots of differential localization probabilities when using the informative prior. (right)

The differential localisation probabilities in both scenarios plotted against each other. Principal component regression overlaid suggests that the default prior has generally higher differential localisation probabilities.

To alleviate confusion in the methods sections we have added:

As a default, we set $\alpha_{jj} = 1.01$ and $\alpha_{jk} = 0.01$ for $k \neq j$. This assumes that there are roughly an order of magnitude fewer differentially localised proteins than spatially stable ones. This default was used in all simulations and applications except the EGF simulation dataset. In that case, we had prior knowledge of a differential localisation between the plasma membrane and the endosome and so we set the corresponding entry of α to 1 .

To avoid too much additional text in the EGF results section, we have added the following to clarify the informative prior was used in the EGF application:

... with an informative prior (see methods). Sensitivity to these prior choices are assessed in the supplementary materials (appendix 11)

- The expected calibration error does not seem to be the best metric to show accuracy of the probabilities. For example, it could simply be dominated by the non-translocated predictions, i.e. what is the ECE if one just predicts zero translocation probability for all proteins? If one sticks with the ECE, it could be clarified by plots showing the calibration error for the different bins. However, as Reviewer #2 already seems to allude to, one can also sort the proteins by localization posterior error probability (PEP) and compute the FDR as the average of the PEPs (<https://doi.org/10.1021/pr700739d>). This FDR can then be compared to the ground truth in the simulation studies and should ideally be shown to produce conservative estimates. I acknowledge that this might be a too high standard to ask for and the number of translocations

is too small to produce smooth FDR estimates. However, such an analysis would at least give users a rough idea how much they can trust the probability estimates.

This is an important question from the reviewer and we agree it would be a good idea to give a better understanding of the calibration of the differential localisation probabilities. We have thus performed the simulations and analysis suggested by the reviewer. We determined whether BANDLE can provide good estimates for the proportion of false discoveries amongst those flagged as differentially localized. Due to fluctuations, we would expect these results to hold for an ensemble of datasets but due to computational limitations we limited the analysis to 10 simulated datasets. In the proceeding text, which has been added to the supplementary material, we find that BANDLE generally tends to be anti-conservative, although at the 1% level BANDLE provides the correct nominal control of the FDP (false discovery proportion). Furthermore, we noted that the false positives, that causes BANDLE to become uncalibrated, are generally outliers. By filtering out proteins that BANDLE deemed to be outliers ($p(\phi_i = 1) > 0$), the calibration is improved significantly and BANDLE tends to control the FDP at all levels of practical interest.

We have added the following text to the supplementary material (1.21.19).

Expected false discovery proportion

Here, we continue the line of analysis discussed in the previous section about how to interpret the differential localisation probabilities. One desirable property would be if the differential localisation probabilities provided an estimate of the false discovery proportion (FDP). In the present context, the FDP is the proportion of proteins called as differentially localised that are in fact not differentially localised, and we would like to be able to control this quantity at a prespecified level. For example, if 100 proteins are called differentially localised, we would like only $x\%$ of these to be false discoveries. In practice, we do not know which are the negative/positive cases and so the FDP must be estimated. Letting $\hat{\chi}_i$ be an estimate of the differential localisation probability and then assuming that these are sorted in decreasing order, the estimated FDP for a collection of n proteins is given by:

$$\widehat{\text{FDP}}(n) = \frac{1}{n} \sum_{i=1}^n (1 - \hat{\chi}_i).$$

This can be interpreted as the average posterior error probability (PEP) (Käll et al., 2008). Ideally, we would like the predicted FDP to match the observed FDP. Alternatively, it would also be acceptable if the predicted FDP were to overestimate the observed FDP (i.e. it is conservative), since controlling the predicted FDP would still also control the observed FDP in this case. To assess this for BANDLE we generate 10 simulated datasets as in the previous section. Initially, we ignore the information provided by BANDLE regarding which proteins are identified as outliers. We report plots of the predicted FDP against the expected FDP (see Figure 57). In this case, we observe that the BANDLE differential localisation probabilities generally tend to be anti-conservative; i.e., BANDLE generally tends to overestimate the

probability of a protein being differentially localised (although we note that, at a 1% level, the differential localisation probabilities provide the correct nominal control of the FDP). Although this might at first seem concerning, it is important to reiterate that we have so far ignored the information that BANDLE provides regarding which proteins are identified as outliers (which, in general, we would not advise). As a consequence, outlier proteins are being incorrectly called as differentially localised. If we instead use BANDLE as intended by first filtering out outliers (i.e., those proteins for which $p(\phi_i = 1) > 0$), and then recompute the calibration plots, then we observe high-quality calibration of the differential localisation probabilities (see Figure 58). For transparency, in Figure 59, we zoom in on the results up to the 0.05 level, where we can see that we have generally very good calibration, while acknowledging that in any given dataset we might be very slightly conservative or anti-conservative.

To direct the readers to this important point from the main text we have added:

In the supplementary material (section 1.21.19), we demonstrate that the differential localisation probabilities provide estimates of the FDR at a 1% level and at all levels once outliers are filtered.

Minor comments:

- In section 4.2.5, the re-use of the rho variable is potentially confusing with the hyperparameter rho in the previous section.

Indeed, we agree with the reviewer about the potential confusion and have switched to using gamma

- The term “feature” appears several times without explicitly explaining what they refer to.

This is an important clarification. The features in the case of spatial proteomics are the subcellular fractions on which the MS measurements are taken. We have clarified in the first appearance:

Features, i.e. subcellular fraction, ...

- When introducing Gaussian Random Fields on page 4, it would be helpful to already mention its connection to Gaussian Processes which is a much more widely known concept. Without being too familiar with the exact terminology, I also don't understand the need for calling it a Gaussian Random Field, even if one is being “pedantic” as the authors put it. It seems that the time dimension is never modeled with a Gaussian Process, so calling it a Gaussian Random Field made it more confusing for me, as I was searching for that extra dimension.

The reviewer raises an important point. Whilst presenting this work, we originally refer to them as Gaussian processes which caused multiple confusions. Firstly, some audiences were confused about the lack of temporal dimension in the data. Secondly, some audiences were confused by the connection with Gaussian process classification. Referring to them as Gaussian random fields more readily connected people with spatial modelling, despite, as the reviewer points out, there only being a single dimension.

When introduced on page 4, we have added after the Gaussian random field is introduced:

which is also frequently referred to as a Gaussian Process

- Figure 1d: can a differential localization probability of ~ 0.4 really be considered stable?

This is referring to Protein C which is spatially stable by definition. 0.4 may be a high probability for this but is simply a cartoon to give an idea of the model outputs.

- Figure 3a: it is unclear which translocations we should expect as all translocated proteins have the same marker.

This figure has been updated to use letter markers to make it clear which proteins match across the two plots.

- Figure 5: mix of uniprot IDs and gene names is confusing

We agree with the reviewer that this is confusing. We have reverted to uniprot IDs because it is more consistent and is more amenable to programmatic analysis.

- The authors mention several times that they only looked at two conditions for the sake of brevity, but can BUNDLE actually handle more than two conditions? To me, it's not completely trivial how one would choose and evaluate priors in this case.

No, BUNDLE cannot explicitly handle more than two conditions within one integrated modeling framework, although it could be applied pairwise to each of the conditions if there were more than two. One would need to be careful with respect to prior choice and FDR control, however. Modeling more than one condition would be complicated, since the α matrix would need to be structured carefully. It is likely some sort of Markov-structure may need to be imposed, which is work beyond this manuscript. We have slightly edited the text in the discussion to clarify:

Furthermore, our approach here can only look at pairs of conditions at a time. In the future, more complex spatial proteomics designs will be available that will study multiple perturbations simultaneously and our approach will be adapted accordingly.

Software comments:

- In the documentation of fitGP, there is no explanation of how to choose a prior for the hyppar parameter (prior, length-scale amplitude and standard deviation). It shouldn't be necessary to look into the Crook et al. 2021 publication.

We have added to the documentation of fitGP function an explanation of the hyppar parameter and some guidance on how to check the fit of the GPs. This has been pushed directly to the current development version of Bioconductor and will appear in versions > 1.1.3. Specifically, we have added to the documentation pages:

This set of functions allow users to fit GPs to their data. The fitGPmaternPC function allows users to pass a vector of penalised complexity hyperparameters using the hyppar argument. You must provide a matrix with 3 columns and 1 row. The order of these 3 columns represent the hyperparameters length-scale, amplitude, variance. We have found that the matrix(c(10, 60, 250), nrow = 1) worked well for the spatial proteomics datasets tested in Crook et al (2021). This was visually assessed by passing these values and visualising the GP fit using the plotGPmatern function (please see vignette for an example of the output). Generally, (1) increasing the lengthscale parameter (the first column of the hyppar matrix) increases the spread of the covariance i.e. the similarity between points, (2) increasing the amplitude

parameter (the second column of the hyppar matrix) increases the maximum value of the covariance and lastly (3) decreasing the variance (third column of the hyppar matrix) reduces the smoothness of the function to allow for local variations. We recommend users start with the recommended parameters and change and assess them as necessary for their dataset by visually evaluating the fit of the GPs using the `plotGPmatern` function. Please see the vignettes for more details and examples.

- For the authors' information, I had to install `libxml2-dev`, `zlib1g-dev`, `libnetcdf-dev` to install BUNDLE in an R Docker instance, perhaps this can be documented somewhere as the R error and warning messages are not the most helpful.

We thank the reviewer for highlighting this and have added a note to the package README file on the Github landing page under "Installation troubleshooting".

- In vignette 1, section 6.5, the authors set an 1% FDR threshold on the differentially localized proteins, but then simply use a cutoff of 0.99. In reality, the FDR is the average error probability of the ranked proteins.

We have corrected this in the vignette.

Spelling/Grammar:

- Page 18 endsomes -> endosomes
- Page 25 and those that are not -> than those that are not
- Page 26 cell states -> cell state; those were -> those where ... was
- Page 28 dependant -> dependent?
- Page 28 is obtain -> is obtained; be interpret -> been interpreted

Thank you for spotting these typos, they have now been corrected. Indeed, dependent is the correct form in this case.

REVIEWERS' COMMENTS

Reviewer #3 (Remarks to the Author):

The authors have sufficiently addressed my remaining concerns.

Minor comments:

- Update the legend of Figure 3A to state that the simulated translocations are indicated in black letters.
- Supplementary Figure 26 shows the PCA for the control instead of the stimulation.
- Supplementary Figure 57: produce -> produced

To the reviewer,

Reviewer #3 (Remarks to the Author):

The authors have sufficiently addressed my remaining concerns.

Minor comments:

- Update the legend of Figure 3A to state that the simulated translocations are indicated in black letters.
- Supplementary Figure 26 shows the PCA for the control instead of the stimulation.
- Supplementary Figure 57: produce -> produced

We thank the reviewer for providing feedback on the manuscript and we are pleased we have addressed their concerns. The minor changes to the manuscript and supplementary figures have been made.